# MLOT: Extending the Bipartite Structure towards Multi-Layered Structure for Optimal Transport

## Abstract

Despite its remarkable success and widespread adoption in various domains, optimal transport (OT) has a rather simple structure, relying on bipartite graphs with only two layers of nodes for transportation. In this paper, we propose a multi-layered OT approach that extends the original two-layer structure to handle transportation problems across multiple hierarchical levels. Within this framework, the source distribution flows through intermediate layers, before reaching the target distribution. Unlike previous variants of OT that involve multiple distributions, our multi-layered OT typically involves uncertain intermediate distributions, which need to be computed based on the relationships between the preceding and succeeding distributions. Under entropic regularization, MLOT-Sinkhorn algorithm is further proposed for multi-layered OT, which can be accelerated using GPUs and significantly outperforms general solvers such as Gurobi. The theoretical results of our entropic MLOT are also given in this paper. In the experiments, we validate its speed advantage and convergence performance. We further validate its feasibility through Text-Image retrieval and intermediate image computing task, which demonstrates reformulating the problems as MLOT can achieve better results. Source code will be made available.

## 1 Introduction

Optimal Transport (OT) (Peyre & Cuturi, 2019) has been an increasingly important mathematical tool for solving various machine learning problems, with success in a wide range of applications, ranging from domain adaptation (Tzeng et al., 2017), learning generative models (Arjovsky et al., 2017), network designing (Xu & Cheng, 2023), self-supervised contrastive learning (Caron et al., 2020), to long-tail recognition (Peng et al., 2021) etc. It allows for the comparison of probability distributions, combining the underlying geometric structure of the sample space.

However, real-world transportation (Bektaş et al., 2019) scenarios are inherently complex, which previous simple transportation often failed to capture. As shown in Fig. 1, we take an example in cross-border e-commerce operations, considering a scenario involving Amazon and FedEx. The source and target distributions are fixed: Amazon's warehouses in different regions (source) have a known distribution of product availability, and the demand from customers (target) is also pre-determined based on market forecasts. However, the intermediate distributions, such as the logistics flow through various transit points (e.g. ports, FedEx sorting hubs) are uncertain and need to be optimized. In this context, the transportation problem transitions from a two-layer network to a multi-layerd one, which also motivates us to delve into the theory of optimal transport within a multi-layered framework.

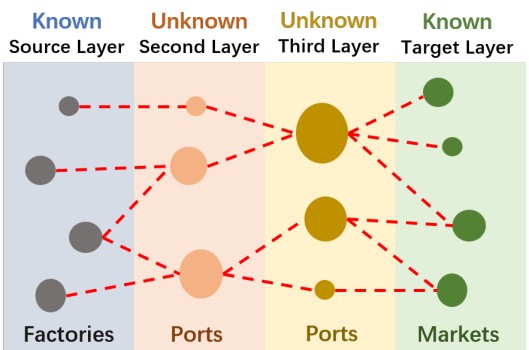

Figure 1: MLOT scenario: mass are transported among several unknown intermediates, aiming to minimize total cost on a geometric distance.

In this paper, we propose a new variant of optimal transport called multi-layered optimal transport (MLOT) that extends the original two-layered transportation structure to multi-layered case. As shown in Fig.1, we assume the known source and target distributions in the source and target layers, along with the known cost matrices between layers. Our objective is to determine intermediate distributions and the transportation plan (i.e., coupling) between layers. Similar to vanilla OT, this problem fundamentally boils down to a linear programming problem (Dantzig, 2002) and one can employ the network simplex method (Grigoriadis, 1986) to solve it, although it proves to be inefficient. Building on prior work (Cuturi, 2013), we endeavor to accelerate the solution of MLOT using matrix iteration algorithms for GPU acceleration.

To achieve fast computation and obtain an approximate solution, we apply entropic regularization to MLOT. The MLOT-Sinkhorn algorithm is proposed through alternating iterations of scaling variables (Cuturi, 2013) and intermediate distributions. Theoretical results for our MLOT are also presented, including the global convergence of our MLOT-Sinkhorn algorithm. We first do the experiments with a small enough coefficient for entropic regularization. The results demonstrate that our MLOT-Sinkhorn algorithm can achieve an objective function close to the solution obtained from Gurobi, but with speeds several tens to hundreds of times faster for larger problem sizes. Furthermore, we view zero-shot retrieval based on CLIP (Radford et al., 2021) as a transportation problem and utilize MLOT to enhance inference through data augmentation. Specifically, we consider the first layer as features of query images, the second layer as features of the text to be retrieved (i.e., captions), and the third layer as features of the augmented images in the first layer. We employ the MLOT-Sinkhorn algorithm for solving this, and experimental results confirm that this inference method has significantly improved compared to previous softmax-based methods without requiring additional training. Besides, based on the calculation of intermediate distributions, we conducted image interpolation experiments. The results indicate that the interpolated images generated using MLOT are relatively clear, serving as a viable alternative method for barycentric interpolation. Finally, **this paper contributes:**

1) We propose MLOT, where we extend the traditional bipartite graph to a multi-layer structure. Source marginals transport mass to uncertain immediate marginals and then further transport the mass to the target marginal.

2) Entropic regularization is applied to MLOT, and the MLOT-Sinkhorn algorithm is derived to obtain an approximate solution for MLOT. Experimental results demonstrate that MLOT-Sinkhorn achieves a solution close to the linear programming solution computed by Gurobi while significantly outperforming Gurobi in terms of computation speed.

3) We present a novel method to convert Zero-shot Text-Image retrieval tasks into MLOT problems using augmented data. This transformation improves retrieval accuracy by effectively utilizing multi-layer information.

4) Building upon the calculation of intermediate distributions, we applied MLOT to image interpolation computations. Experimental results demonstrate that the intermediate images produced by MLOT-Sinkhorn are relatively clear, providing a promising alternative for barycentric interpolation.

## 2 PRELIMINARIES AND RELATED WORK

**Entropic Optimal Transport.** The OT theory can be traced back to (Monge, 1781) where the objective is to seek a mapping that minimizes the total cost of transporting mass from a source measure to a target measure. Kantorovich (Kantorovich, 1942) introduces the idea of using probabilistic transport instead of a deterministic map, which is now commonly known as Kantorovich's formulation of OT. Specifically, given the cost matrix $\mathbf{C} \in \mathbb{R}_{m \times n}^+$ and two histograms $(\mathbf{a}, \mathbf{b})$ where $n$ and $m$ are numbers of dimensions, Kantorovich's OT involves solving the coupling $\mathbf{P}$ (i.e. the joint probability matrix):

$$\min_{\mathbf{P} \in U(\mathbf{a}, \mathbf{b})} <\mathbf{C}, \mathbf{P}> \quad \text{where} \quad U(\mathbf{a}, \mathbf{b}) = \{\mathbf{P} \in R_{mn}^+ | \mathbf{P}\mathbf{1}_n = \mathbf{a}, \mathbf{P}^\top \mathbf{1}_m = \mathbf{b}\}. \tag{1}$$

Relaxing with the entropic regularization (Wilson, 1969) is one of the simple yet efficient methods for solving OT, which can be formulated as:

$$\min_{\mathbf{P} \in U(\mathbf{a}, \mathbf{b})} <\mathbf{C}, \mathbf{P}> -\epsilon H(\mathbf{P}), \tag{2}$$

where $\epsilon > 0$ is the coefficient for entropic regularization $H(\mathbf{P})$, and $H(\mathbf{P})$ can be specified as $H(\mathbf{P}) = - < \mathbf{P}, \log \mathbf{P} - \mathbf{1}_{m \times n} >$. The objective in Eq. 2 is $\epsilon$-strongly convex, and thus it has a unique solution, which satisfies $\mathbf{P}_\epsilon^* = \mathrm{diag}(\mathbf{u})\mathbf{K}\mathrm{diag}(\mathbf{v})$, where $\mathbf{K} = e^{-\mathbf{C}/\epsilon}$ is the Gibbs kernel associated to the cost matrix $\mathbf{C}$ and $(\mathbf{u}, \mathbf{v})$ are two (unknown) scaling variables (Cuturi, 2013).

**Optimal Transport with Multiple Marginals.** Instead of coupling two histograms $(\mathbf{a}, \mathbf{b})$ in Kantorovich problem (Kantorovich, 1942), the multi-marginal optimal transportation (Abraham et al., 2017) couples $K$ histograms $(\mathbf{a}_k)_{k=1}^K$ by solving the following multi-marginal transport:

$$\min_{\mathbf{P} \in U((\mathbf{a}_k)_k)} < \mathbf{C}, \mathbf{P} >= \sum_k \sum_{i_k=1}^{n_k} \mathbf{C}_{i_1, i_2, \ldots, i_K} \mathbf{P}_{i_1, i_2, \ldots, i_K} \tag{3}$$

where $\mathbf{C}_{i_1, i_2, \ldots, i_K}$ is $n_1 \times \cdots \times n_K$ cost tensor and the valid coupling set $U((\mathbf{a}_k)_{k=1}^K)$ is defined as

$$U((\mathbf{a}_k)_k) = \{\mathbf{P} \in \mathbb{R}_{n_1 \times n_2 \ldots n_K}^+ | \forall k, \forall i_k, \sum_{l \neq k} \sum_{i_l=1}^{n_l} \mathbf{P}_{i_1, \ldots, i_K} = \mathbf{a}_{k, i_k}\}. \tag{4}$$

Note the Multi-Marginal Optimal Transport has various applications including image processing (Rabin et al., 2012), financial mathematics for derivative pricing (Galichon et al., 2014) and so on (Pass, 2015). Compared with MLOT, the Multi-Marginal Optimal Transport approach differs in that all of its marginals are deterministic, and its objective is to compute the coupling tensor between multiple marginals, rather than the coupling between two marginals in this paper.

**Optimal Transport on a Graph.** The optimal transport on graphs can be traced back to (Feldman & McCann, 2002), which first calculates the shortest distances between source nodes and target nodes to create a cost matrix, subsequently using it to compute the 1-Wasserstein distance. This approach transforms the problem into a linear program, and more precisely, a min-cost flow problem, which has been utilized and extended to define and study traffic congestion models. Recently, (Le et al., 2022) introduced a new variant called Sobolev transport (ST), designed for measures supported on graphs, which allows for a closed-form expression for faster computation. Additionally, (Le et al., 2024) generalized Sobolev transport with an Orlicz structure (Orlicz, 1932). However, the above works rely on the calculating the shortest distances on graph firstly, so they do not directly compute the transport couplings in the graph. In this paper, we attempt to directly compute the transportation between nodes in a multi-layer structure. We propose an algorithm that can compute the optimal flow as well as intermediate distributions directly based on ground metric, no need for shortest path on graph.

# 3 METHODOLOGY

## 3.1 MULTI-LAYERED OPTIMAL TRANSPORT

**Formulation.** We first give the definition of our Multi-Layered Optimal Transport (MLOT). Given the known source distribution $\mathbf{a}_1$ and target distribution $\mathbf{a}_K$, our MLOT aims to transport the source distribution through intermediate uncertain distributions $(\mathbf{a}_2, \mathbf{a}_3, \ldots, \mathbf{a}_{K-1})$ to the target distribution $\mathbf{a}_K$, where $\mathbf{C}_k \in \mathbb{R}_{n_k \times n_{k+1}}^+$ is known as the cost matrix between $\mathbf{a}_k$ and $\mathbf{a}_{k+1}$. Our goal is to solve for the optimal couplings $(\mathbf{P}_k)_{k=1}^{K-1}$ and the intermediate distributions $(\mathbf{a}_k)_{k=2}^{K-1}$ with the following optimization:

$$\min_{(\mathbf{P}_k)_k, (\mathbf{a}_k)_k} \sum_{k=1}^{K-1} < \mathbf{C}_k, \mathbf{P}_k > \quad \text{s.t.} \quad \mathbf{P}_k \mathbf{1}_{n_{k+1}} = \mathbf{a}_k, \quad \text{and} \quad \mathbf{P}_k^\top \mathbf{1}_{n_k} = \mathbf{a}_{k+1}, \forall k = 1, \ldots, K-1. \tag{5}$$

Note that when $K = 2$, our MLOT degenerates to the original Kantorovich OT as proposed in Eq. 1. One efficient way to solve the above problem is through Graph OT methods based on the shortest path algorithm, as proposed by (Titouan et al., 2019), where the shortest path distances between source and target nodes are first computed, followed by a heuristic algorithm to determine the final solution. However, such algorithms do not directly involve the computation of intermediate distributions $(\mathbf{a}_k)_{k=2}^{K-1}$, limiting their applicability in real-world scenarios. For instance, in the cross-border e-commerce operations problem mentioned in the introduction, if we introduce capacity constraints for goods transportation at ports, which are indeed present in real scenarios and need to be considered, the original shortest path-based algorithms become impractical.

**Relation to Wasserstein Barycenter.** We found that our MLOT can be linked to the Wasserstein barycenter. When considering the distributions $(\mathbf{b}_s)_{s=1}^S$, the Wasserstein barycenter among them aims to learn the distribution $\mathbf{a}$ which optimizes:

$$\min_{(\mathbf{P}_s)_s, \mathbf{s}} \sum_{s=1}^S \lambda_s < \mathbf{D}_s, \mathbf{P}_s > \quad \text{s.t.} \quad \mathbf{P}_s \mathbf{1} = \mathbf{b}_s, \quad \mathbf{P}_s \mathbf{1} = \mathbf{a} \quad \forall s = 1, 2, \ldots, S \tag{6}$$

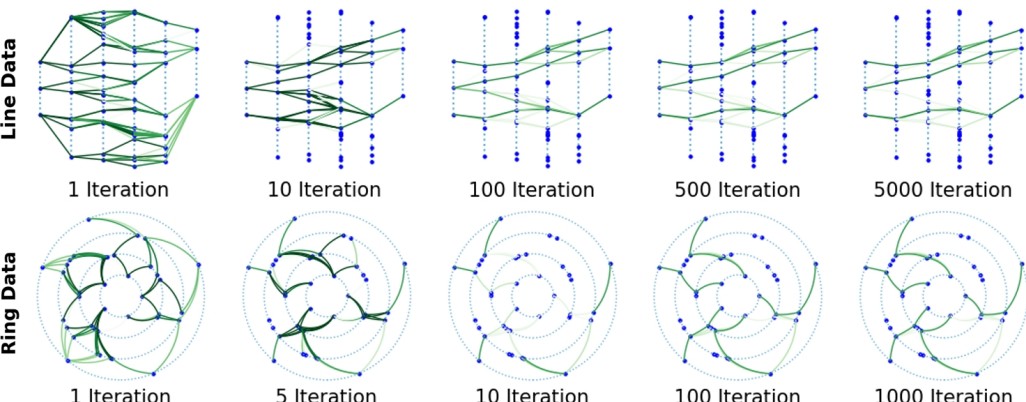

Figure 2: Transportation results of MLOT on synthetic Line and Ring data (refer to the data setup in Section 4) and the thickness of the green line is directly proportional to the value of transportation. By varying the iterations, couplings become sharper, and eventually converge to optimal transportation of entropic MLOT.

where $\mathbf{D}_s$ is the distance matrix between $\mathbf{a}$ and $\mathbf{b}_s$. As mentioned in MLOT formulation, our MLOT assumes that the source and target distributions are known, and the objective is to compute the intermediate distributions. In contrast, the Wasserstein barycenter assumes that one or several target distributions of the transportation are known, and the goal is to compute the source distribution. Specifically, when $S = 2$ in Eq.6 and $K = 3$ in Eq.5, the optimization of our MLOT is equivalent to solving the Wasserstein barycenter by setting $\mathbf{C}_1 = \mathbf{D}_1^\top$ and $\mathbf{C}_2 = \mathbf{D}_2$. In this paper, following (Cuturi, 2013), we consider MLOT under entropic regularization in the next subsection, where we directly compute the coupling between each pair of layers and intermediate distributions instead of relying on indirect calculations through shortest paths.

## 3.2 MLOT WITH ENTROPIC REGULARIZATION AND MLOT-SINKHORN ALGORITHM.

In this subsection, we attempt to introduce entropy regularization to MLOT in order to obtain a GPU-friendly Sinkhorn-like algorithm, which can iteratively compute an approximate solution for MLOT through matrix iterations. Unlike the case of vanilla OT, MLOT not only requires optimizing coupling $\mathbf{P}_k$ but also involves considering intermediate distribution $\mathbf{a}_k$. Here, we contemplate applying entropy regularization to both, leading to the formulation of entropic MLOT as:

$$\min_{(\mathbf{P}_k)_k, (\mathbf{a}_k)_k} \sum_{k=1}^{K-1} \Big( <\mathbf{C}_k, \mathbf{P}_k> -\epsilon H(\mathbf{P}_k) \Big) - \tau \sum_{k=2}^{K-1} H(\mathbf{a}_k) \quad \text{s.t.} \quad \forall k, \mathbf{P}_k \mathbf{1}_{n_{k+1}} = \mathbf{a}_k, \mathbf{P}_k^\top \mathbf{1}_{n_k} = \mathbf{a}_{k+1},$$

(7)

where $\epsilon > 0$ and $\tau > 0$ are coefficients for the regularization terms $H(\mathbf{P}_k)$ and $H(\mathbf{a}_k)$, respectively. The optimization described above is essentially a convex optimization problem, ensuring the existence of a unique optimal solution. In particular, as $\epsilon \to 0$ and $\tau \to 0$, the entropic MLOT in Eq. 7 degenerates to the original MLOT in Eq. 5. Furthermore, we can further derive properties of the solution as follows by using the method of Lagrange multipliers.

**Proposition 1** (Solution Form). *The solutions to Eq. 7 is unique and the solution of couplings have the following form for $k = 1, \ldots, K - 1$:*

$$\mathbf{P}_k = Diag(\mathbf{u}_k) \mathbf{S}_k Diag(\mathbf{v}_k) \tag{8}$$

*where $\mathbf{S}_k \{(u_k, v_k)\}_k$ are the set of unknown scaling variables. While the solution of the intermediate distributions satisfying following equations for $k = 2, 3, \ldots, K - 1$:*

$$\mathbf{a}_k = \begin{cases} (\mathbf{u}_k \odot \mathbf{v}_{k-1})^{-\epsilon/\tau} & \tau > 0 \\ \left( (\mathbf{S}_{k-1}^\top \mathbf{u}_{k-1}) \odot (\mathbf{S}_k \mathbf{v}_k) \right)^{1/2} & \tau = 0 \end{cases} \tag{9}$$

The proof are given in Appendix B. Compared to entropic OT, the coupling form of MLOT is similar, both expressed as the product of the Gibbs kernel $\mathbf{S}_k$ and two diagonal matrices. The difference lies in the fact that our MLOT requires further computation of intermediate distributions as shown in Eq. 9, which implies that the matrix iteration algorithm corresponding to it is inevitably more complex than the Sinkhorn algorithm based on Entropic OT.

**Proposition 2.** *Redefine a general KL divergence as*

$$\widetilde{KL}(\mathbf{P}|\mathbf{S}) = \sum_{ij} \mathbf{P}_{ij} \log \frac{\mathbf{P}_{ij}}{\mathbf{S}_{ij}} - \mathbf{P}_{ij} + \mathbf{S}_{ij}, \tag{10}$$

Figure 3: Impact visualization of $\varepsilon$ on the MLOT-Sinkhorn algorithm solutions, generated by varying $\varepsilon = 8 \times 10^{-2}, 8 \times 10^{-3}, 8 \times 10^{-4}$, and 0 (Gurobi) with $\tau = 0$. The experiments is conducted on Line data. As $\varepsilon$ decreases, the solution of our algorithm progressively converges towards the exact solution of Eq. 5.

*the optimization in Eq. 7 is equivalent to the following minimization, where $(\mathbf{S}_k)_{ij} = e^{-(\mathbf{C}_k)_{ij}/\epsilon}$, and $\mathbf{\Delta}_k = \mathbf{1}_{n_k}/n_k$ represents uniform distribution:*

$$\min_{(\mathbf{P}_k)_k, (\mathbf{a}_k)_k} \varepsilon \sum_{k=1}^{K-1} \widetilde{KL}(\mathbf{P}_k | \mathbf{S}_k) + \tau \sum_{k=2}^{K-1} KL(\mathbf{a}_k | \mathbf{\Delta}_k). \tag{11}$$

The proof is given in Appendix C. Prop. 2 shows the optimal solutions $(\mathbf{P}_k)_k, (\mathbf{a}_k)_k$ exactly minimize the weighted summation of two KL divergence. Compared to entropic OT, the KL projection of $\mathbf{P}_k$ is similar. The difference lies in two places, one is that MLOT includes the summation of KL divergence for all layers, the other is that MLOT also contains the KL projection of the intermediate layers. Expect for the different form between entropic OT and MLOT, they share the similar static Schrödinger form, i.e. the optimization on KL projection. Therefore many methods in entropic OT can be applied to MLOT, such as Bregman Sinkhorn.

**Proposition 3** (Convergence with $\varepsilon$ and $\tau$). *When regularization on intermediate is cancelled ($\tau = 0$), the unique solution $(\mathbf{P}_k^{\varepsilon, \tau})_k$ of Eq. 7 converges to the optimal solution $\mathbf{P}_k^{\star}$ of Eq. 5, as $\varepsilon \to 0$.*

$$(\mathbf{P}_k^{\varepsilon, 0})_k \xrightarrow{\varepsilon \to 0} \arg\min_{(\mathbf{P}_k)_k} \sum_{k=1}^{K-1} <\mathbf{C}_k, \mathbf{P}_k>. \tag{12}$$

*When intermediate is regularized by $\tau$, given fixed $\varepsilon = \varepsilon_0$, the unique solution $(\mathbf{P}_k^{\varepsilon_0, \tau})_k$ of Eq. 7 converges to $(\mathbf{P}_k^{\varepsilon_0, 0})_k$ as $\tau \to 0$.*

$$(\mathbf{P}_k^{\varepsilon_0, \tau})_k \xrightarrow{\tau \to 0} \arg\min_{(\mathbf{P}_k)_k} \sum_{k=1}^{K-1} <\mathbf{C}_k, \mathbf{P}_k> -\varepsilon_0 H(\mathbf{P}_k). \tag{13}$$

The proof is in Appendix. D. Prop. 3 is essentially due to the fact that entropic regularization is a continuous function. This property demonstrates good convergence of MLOT. Eq. 12 and Eq. 13 show respectively that the regularization problem converges to the non-regularization case for both couplings and intermediate. Fig. 3 and Fig. 4 show visually the effect of these two convergences.

**MLOT-Sinkhorn Algorithm.** Next, we delve into algorithm design for solving the entropic MLOT, which is GPU-friendly and hence accelerates the approximation of the optimal solution of MLOT. Based on the above Proposition, here we propose MLOT-Sinkhorn algorithm, the Sinkhorn-like iterative method for calculating the optimal solution of Eq. 7 via matrix-vector iterations. To get the results, an intuitive idea is to iteratively update the coupling $\mathbf{P}_k$ and intermediate distributions $\mathbf{a}_k$ until convergence. Thus for updating the coupling $\mathbf{P}_k$, based on the solution form $\mathbf{P}_k = \text{diag}(\mathbf{u}_k)\mathbf{S}\text{diag}(\mathbf{v}_k)$ and the marginal constraints (i.e. $\mathbf{P}_k\mathbf{1}_{n_{k+1}} = \mathbf{a}_k$ and $\mathbf{P}_k^{\top}\mathbf{1}_{n_k} = \mathbf{a}_{k+1}$), we derive the following iterations for $\mathbf{u}_k^{(l)}$ and $\mathbf{v}_k^{(l)}$ given the iteration number $l$:

---

**Algorithm 1:** MLOT-Sinkhorn Algorithm

**Input** : Source distribution $\mathbf{a}_1$, target distribution $\mathbf{a}_K$, distance metrics $(\mathbf{C}_k)_k, \varepsilon, \tau$

**Output:** Couplings $(\mathbf{P}_k)_k$ and intermediate distributions $(\mathbf{a}_k)_k$

**Initialize** $\mathbf{S}_k = \exp(-\mathbf{C}_k/\varepsilon), \mathbf{u}_k = \mathbf{1}, \mathbf{v}_k = \mathbf{1}$ for $\forall k = 1, 2, \dots K-1$ and $\mathbf{a}_k = \mathbf{1}/N_k$ for $\forall k = 2, 3, \dots K-1$;

**while** *not Converge* **do**
  **for** $k = 1, 2, \dots, K-1$ **do**
    $\mathbf{u}_k \leftarrow \mathbf{a}_k \oslash \mathbf{S}_k\mathbf{v}_k$;
    $\mathbf{v}_k \leftarrow \mathbf{a}_{k+1} \oslash \mathbf{S}_k^{\top}\mathbf{u}_k$;
    **if** $k > 1$ **then**
      Update $\mathbf{a}_k$ via Eq. 15;
    **end**
  **end**
**end**
Calculate $\mathbf{P}_k \leftarrow \text{Diag}(\mathbf{u}_k)\mathbf{S}_k\text{Diag}(\mathbf{v}_k)$ for $\forall k = 1, 2, \dots K-1$;
**return** $(\mathbf{P}_k)_k$ and $(\mathbf{a}_k)_k$;

---

$$\mathbf{u}_k^{(l+1)} = \frac{\mathbf{a}_k^{(l)}}{\mathbf{S}_k\mathbf{v}_k^{(l)}}, \quad \text{and} \quad \mathbf{v}_k^{(l+1)} = \frac{\mathbf{a}_{k+1}^{(l)}}{\mathbf{S}_k^{\top}\mathbf{u}_k^{(l+1)}}, \tag{14}$$

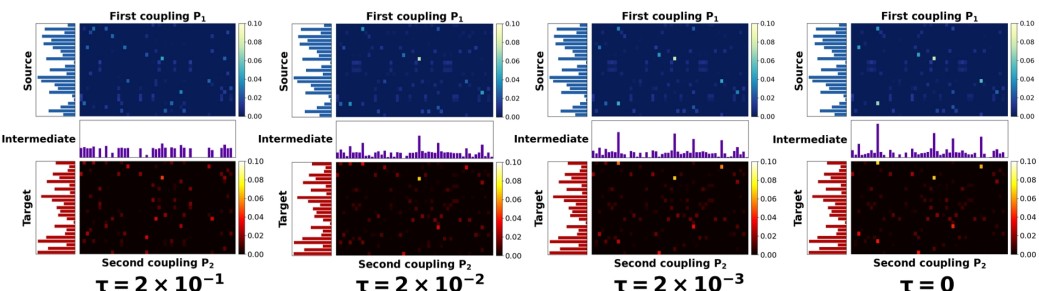

Figure 4: Impact visualization of $\tau$ on the MLOT-Sinkhorn. The experiment is conducted on Line data, by fixed $\varepsilon = 1 \times 10^{-3}$ and varying $\tau = 2 \times 10^{-1}, 2 \times 10^{-2}, 2 \times 10^{-3}$, and 0 (without regularization on intermediate). As $\tau$ decreases, the solution progressively converges towards the solution without regularization on intermediate.

where initialization is set as $\mathbf{v}_k = \mathbf{1}_{n_k}$ and $\mathbf{a}_k = \mathbf{1}/N_k$. Furthermore, for the iteration of the immediate distribution, due to Eq. 9 in Prop. 1, we can update $\mathbf{a}^{(l+1)}$ via

$$\mathbf{a}_k^{(l+1)} = \begin{cases} \left(\mathbf{u}_k^{(l+1)} \odot \mathbf{v}_{k-1}^{(l+1)}\right)^{-\epsilon/\tau} & \tau > 0 \\ \left((\mathbf{S}_{k-1}^\top \mathbf{u}_{k-1}^{(l+1)}) \odot (\mathbf{S}_k \mathbf{v}_k^{(l+1)})\right)^{1/2} & \tau = 0 \end{cases} \tag{15}$$

for $k = 2, \ldots, K - 1$. Then, we iteratively alternate between solving Eq.14 for the underlying coupling $\mathbf{P}_k$ and Eq.15 for intermediate distributions for all $k$ until convergence. This process allows us to obtain the final solutions $(\mathbf{P}_k)_k$ and $(\mathbf{a}_k)_k$. Note in the limit as $\epsilon \to 0$ and $\tau \to 0$ (or $\tau = 0$), empirical evidence demonstrates that the iterative results of our MLOT-Sinkhorn approach closely approximate the exact solution of MLOT obtained using Gurobi.

**Global Convergence of MLOT-Sinkhorn Algorithm** Then we give the global convergence analysis of MLOT-Sinkhorn iteration, which is greatly simplified using the Hilbert projective metric defined as:

$$d_{\mathcal{H}}\left(\mathbf{u}, \mathbf{u}'\right) \overset{\text{def.}}{=} \log \max_{i,j} \frac{\mathbf{u}_i \mathbf{u}_j'}{\mathbf{u}_j \mathbf{u}_i'}$$

Several important properties of Hilbert metric are studied in Appendix A.1. For solution form $\mathbf{P}_k = \text{Diag}(\mathbf{u}_k)\mathbf{S}_k \text{Diag}(\mathbf{v}_k)$ of MLOT-Sinkhorn's iterations, a proposition was presented as follow.

**Proposition 4.** *For all layers, the worst bound of error of $\mathbf{u}_k^{l+1}$ is guaranteed by:*

$$\begin{aligned} d_H\left(\mathbf{u}_k^{l+1}, \mathbf{u}_k^*\right) &= \mathcal{O}\left[\left(\frac{\gamma^2(\gamma+2)}{2 - 2\gamma^2 - \gamma^3}\right)^l\right] \quad (\textit{for } \tau = 0) \\ d_H\left(\mathbf{u}_k^{l+1}, \mathbf{u}_k^*\right) &= \mathcal{O}\left[\left(\frac{\gamma}{1 - (\varepsilon/\tau)\gamma}\left(\gamma + \frac{2\varepsilon}{\tau}\gamma + \frac{\varepsilon}{\tau}\right)\right)^l\right] \quad (\textit{for } \tau > 0), \end{aligned} \tag{16}$$

*where $\mathbf{u}^*$ is the unique optimal scaling variable, $\mathbf{u}^{l+1}$ is the $(l+1)$-th iteration of the scaling variable, and $\gamma \in [0, 1]$ stands for the maximum contraction radio of $\mathbf{S}_k$:*

$$\gamma = \max_k \lambda(\mathbf{S}_k) \overset{\text{def.}}{=} \sup\left\{\frac{d_H(\mathbf{S}_k\mathbf{y}, \mathbf{S}_k\mathbf{y}')}{d_H(\mathbf{y}, \mathbf{y}')}, \ \mathbf{y}, \mathbf{y}' \in \mathbb{R}_+^n\right\},$$

*which shows that the positive matrix $\mathbf{S}_k$ is a strict contraction on the cone of positive vectors.*

This proposition is proved in Appendix A. The bound for $d_H\left(\mathbf{v}_k^{l+1}, \mathbf{v}_k^*\right)$ follows a similar form as $\mathbf{u}_k$. Eq. 16 implies that given an approximate radio $\delta$, for proper setting of $\varepsilon, \tau$, the MLOT-Sinkhorn algorithm will perform linear convergence to a $\delta$-approximate solution in $\mathcal{O}(\log \delta)$ iterations.

**Relation to the Dynamic OT and Schrödinger bridge** Fundamentally, our MLOT is akin to Dynamic Optimal Transport (Tong et al., 2020) in that both can be seen as calculating the intermediate steps of the entire transport process. The difference lies in the fact that we fix the positions of each layer or the cost matrices between two layers in our MLOT, while in Dynamic OT, the locations are continuous throughout the entire space. The relationship between the Schrödinger bridge (De Bortoli et al., 2021) and our entropic MLOT is similar to the relationship between the aforementioned two OT variants; both can be regarded as special cases in a discrete state. Therefore, our MLOT can offer new perspectives and approximate computations for Dynamic OT and the Schrödinger bridge. More details and discussion can be found in Appendix G.

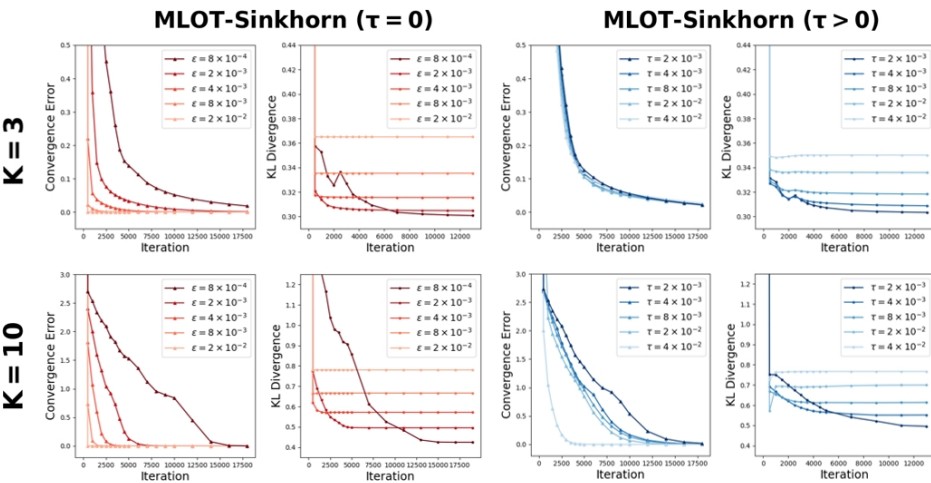

Figure 5: Convergence performance of MLOT on synthetic line dataset, varying 3 layers (top) and 10 layers (bottom). Total points number is 1000, and each layers' number is artificially set. Two indicators are recorded: convergence error and KL difference from ground truth. The left part (red lines) are without regularization on intermidiate, and the right part (blue lines) are with regularization on intermidiate, where $\varepsilon$ is fixed to $1 \times 10^{-3}$. The effect of a gradual decrease in both $\varepsilon$ and $\tau$ leads to more accurate result and slower convergence speed.

## 4 EXPERIMENTS

The experiments of MLOT are conducted on a machine with an NVIDIA GeForce RTX 4090 GPU with 24GB memory. The machine is equipped with an Intel(R) Core(TM) i9-10920X CPU, with a base clock frequency of 3.50GHz. This CPU features 12 cores and 24 threads.

### 4.1 EXPERIMENTS ON SYNTHETIC DATA

In this section, numerical simulation experiments were conducted to validate the efficiency of the MLOT-Sinkhorn algorithm, especially with small values of $\varepsilon$ and $\tau$, as well as to study the convergence performance with respect to these parameters. We first create synthetic datasets by randomly generating a lot of points according to specific multi-layer structure.

**Datasets and Experimental Setting.** We artificially created synthetic datasets as follows. Scenarios of the MLOT problem were modelled with randomly distributed points. The key information of this synthetic dataset includes: Total number of points $N$(Problem size), Number of layers $K$, Number of points per layer $(n_k)_k$, Shape of the layers, and Distance between layers $D$. Based on the shape of the layers, the synthetic dataset can be divided into two parts: Line and Ring. In Line Datasets, points on each layer are distributed on the same straight line with uniform probability, and the lines are parallel to each other. The distance matrix is determined by the Euclidean distance. In Ring Datasets, points on each layer is distributed with uniform probability on a circle, with all circles sharing a common centre. The radius increasing with the index of layers. The distance matrix is determined by the Archimedean spiral length, computed as Appendix E. We adopt Gurobi, a commercial LP solver running on CPUs, as baseline. Additionally, our proposed algorithm is compared to another method that transforms the problem into traditional OT: This process firstly uses shortest-path algorithm (implemented in C++) to transform $K - 1$ distance matrix into a direct matrix from source to target, and then solves it using Python library for traditional OT.

Then we present the results of our experiments conducted on synthetic datasets, which are designed to validate the performance of the MLOT-Sinkhorn algorithm. We first assess its efficiency and running time compared to existing solvers, followed by an evaluation of its convergence performance under various settings.

**Validation of efficiency and running time of MLOT-Sinkhorn.** A visual representation of the Synthetic dataset is shown in Fig. 2. The thickness of the green line is proportional to the value of transportation. Top row: features a Line dataset with $N = 66$, $K = 6$, $(n_k)_k = \{3, 10, 20, 20, 10, 3\}$, $D = 1$, where points in each layer are uniformly distributed along a line of length 6. Bottom row: displays a Ring dataset with $N = 40$, $K = 4$, $(n_k)_k = \{5, 15, 15, 5\}$, $D = 1$, with points uniformly distributed around a circular ring. The MLOT problem is configured with both source and target distributions being uniform, in order to make optimal couplings approach a one-to-one transport. The parameters of MLOT-Sinkhorn are set to $\varepsilon = 1 \times 10^{-3}, \tau = 0$. Fig. 2 illustrates how the solutions returned by MLOT-Sinkhorn evolve with increasing iterations. Since the couplings are initialized as uniform transports, the resulting solutions transition from being even to increasingly

| Problem Size | Gurobi | | Short Path+Sinkhorn | | MLOT($\tau = 0$) | | MLOT($\tau > 0$) | |
|---|---|---|---|---|---|---|---|---|
| | Objective | Time(s) | Objective | Time(s) | Objective | Time(s) | Objective | Time(s) |
| Experiment on synthetic Line data. | | | | | | | | |
| $1 \times 10^2$ | 1.0684 | 0.078 | 1.0692 | 0.168 | 1.0692 | 2.229 | 1.0702 | 1.409 |
| $1 \times 10^3$ | 0.4082 | 6.644 | 0.4099 | 10.249 | 0.4106 | 2.356 | 0.4126 | 1.605 |
| $2 \times 10^3$ | 0.6323 | 43.875 | 0.6336 | 13.342 | 0.6342 | 2.896 | 0.6349 | 1.941 |
| $5 \times 10^3$ | 0.1463 | 329.635 | 0.1487 | 67.815 | 0.1508 | 11.275 | 0.1519 | 7.399 |
| $1 \times 10^4$ | Out Of Memory | | 0.3710 | 421.267 | 0.3707 | 41.154 | 0.3708 | 27.323 |
| $2 \times 10^4$ | Out Of Memory | | 0.1129 | 2575.662 | 0.1137 | 161.594 | 0.1139 | 110.042 |
| Experiment on synthetic Ring data. | | | | | | | | |
| $1 \times 10^2$ | 2.3843 | 0.157 | 2.3848 | 0.339 | 2.3874 | 2.967 | 2.3900 | 2.060 |
| $1 \times 10^3$ | 2.0319 | 20.470 | 2.0341 | 1.242 | 2.0396 | 3.340 | 2.0403 | 2.156 |
| $2 \times 10^3$ | 2.0402 | 45.588 | 2.0427 | 2.723 | 2.0481 | 3.583 | 2.0484 | 2.269 |
| $4 \times 10^3$ | 2.0222 | 323.822 | 2.0249 | 15.324 | 2.0301 | 5.421 | 2.0303 | 3.508 |
| $1 \times 10^4$ | Out Of Memory | | 2.1536 | 336.061 | 2.1588 | 47.382 | 2.1589 | 30.213 |
| $2 \times 10^4$ | Out Of Memory | | 2.1521 | 3125.446 | 2.1573 | 183.724 | 2.1573 | 124.7420 |

Table 1: Experiment on synthetic Line dataset and Ring datasets. The objective and time cost (on seconds) are evaluated by comparing our proposed MLOT-sinkhorn ($\tau = 0$ and $\tau > 0$) with other two baselines. Our proposed algorithms provide highly accurate (avg. error $< 1\%$ in Line data and avg. error $< 1\%$ in Ring data) results in much more efficient time.

sharp, ultimately approaching the scenario of minimal cost. We carefully examined the accuracy and running time of the MLOT-Sinkhorn algorithm when both $\varepsilon$ and $\tau$ are small. Experiments were conducted on both the Line dataset and the Ring dataset, varying the problem size $N$ from $1 \times 10^2$ to $2 \times 10^4$. In the Line dataset, we set $K = 3$, $(n_k)_k = \{N/4, N/2, N/4\}$, $D = 0.1$, with points in each layer uniformly distributed along a line of length 20. For the Ring dataset, we also set $K = 3$, $(n_k)_k = \{N/4, N/2, N/4\}$, $D = 1$, and points in each layer uniformly distributed around the ring. The parameters for two MLOT-Sinkhorn are set as $\epsilon = 1 \times 10^{-3}$, $\tau = 0$ and $\epsilon = 1 \times 10^{-3}$, $\tau = 2 \times 10^{-3}$. The stopping condition is either when the number of iterations exceeds 20000 or when the difference in update is less than $10^{-15}$. The results are shown in Tab. 1. For various problem sizes, the objective values obtained from MLOT are highly consistent with those from the Gurobi solver, with average relative errors being less than 0.7%. In addition to its accuracy, MLOT-Sinkhorn operates several times faster than Gurobi and dynamic programming. As the problem size increases, the memory requirements for the Gurobi solver become prohibitive, leading to "Out of Memory" when problem size reaches $1 \times 10^4$. In contrast, MLOT-Sinkhorn can efficiently handle larger problem sizes while maintaining high speed and accuracy. The results display the efficiency of our algorithm.

**Convergence performance of MLOT-Sinkhorn.** An demonstration of the convergence of MLOT-Sinkhorn is illustrated in Fig. 3 and Fig. 4. The depth of color in the heatmaps indicates the magnitude of the transport values at each location, while the central bar graphs represent the intermediate distributions computed by the algorithm. This experiment aims to showcase the convergence properties regarding $\varepsilon$ and $\tau$ as proven in Prop. 3. The Experiment is conducted on Line dateset, with $N = 100$, $K = 3$, $(n_k)_k = \{25, 50, 25\}$, $D = 5$, where points in each layer are uniformly distributed along a line of length 20. Both the source and target distributions were randomly generated and normalized. In Fig. 3, $\tau$ is set to 0, and a series of decreasing $\varepsilon$ values are employed, comparing to the ground truth solution of Eq. 5 ($\varepsilon = 0$), which illustrate the convergence of MLOT-Sinkhorn with respect to $\varepsilon$. In Fig. 4, $\varepsilon$ is fixed as $1 \times 10^{-3}$, and a series of decreasing $\tau$ values are employed, demonstrating the convergence of MLOT-Sinkhorn with respect to $\tau$.

We further carefully investigate the convergence of the MLOT-Sinkhorn algorithm with respect to $\varepsilon$ and $\tau$. Experiment is conducted on Line dateset with $N = 1000$, $D = 5$, and points in each layer uniformly distributed along a line of length 20. For different layer numbers, we manually set $(n_k)_k$, in order to create a dataset shape that "gradually increases from the source to the intermediate layers, and then gradually decreases to the target". Two indicators that characterize convergence performance are recorded: the convergence error of $(\mathbf{u}_k)_k$ and $(\mathbf{v}_k)_k$, and the KL difference from ground truth of intermediate layer. The results are shown in Fig. 5. Top row: features $K = 3$ with $(n_k)_k = \{250, 500, 250\}$. Bottom row: features $K = 10$ with $(n_k)_k = \{25, 50, 125, 150, 150, 150, 150, 125, 50, 25\}$. The left section (red lines) represents cases without regularization on the intermediate layers, i.e. $\tau = 0$, where $\varepsilon$ decreases from $2 \times 10^{-2}$ to $8 \times 10^{-4}$. From the graph, it is evident that smaller $\varepsilon$ values lead to solutions closer to the ground truth, although they require longer to converge for $(\mathbf{u}_k)_k$, $(\mathbf{v}_k)_k$. The right section (blue lines) incorporates regularization on the intermediate layers, with $\varepsilon$ fixed at $1 \times 10^{-3}$. As $\tau$ decreases from $4 \times 10^{-2}$ to $2 \times 10^{-3}$, the graphs similarly show that smaller $\tau$ values yield solutions closer to the ground truth, though they also require more time to achieve iterative convergence.

Table 2: Comparison of zero-shot retrieval performance between standard softmax inference and our proposed MLOT algorithm on COCO and Flickr30k datasets. Results are presented for two model structures (ViT-B/32 and RN50x64) across both Text-to-Image and Image-to-Text retrieval tasks, measured by R@1, R@5, and R@10 metrics.

| | | COCO | | | | | | Flickr30k | | | | | |
| | | Text⇒Image | | | Image⇒Text | | | Text⇒Image | | | Image⇒Text | | |
| Structure | Inference | R@1 | R@5 | R@10 | R@1 | R@5 | R@10 | R@1 | R@5 | R@10 | R@1 | R@5 | R@10 |
|---|---|---|---|---|---|---|---|---|---|---|---|---|---|
| ViT-B/32 | softmax | 29.02 | 52.84 | 64.26 | 49.82 | 74.64 | 83.10 | 24.42 | 42.96 | 51.00 | 34.34 | 54.44 | 61.97 |
| | **MLOT(Ours)** | 35.10 | 61.22 | 72.18 | 50.66 | 75.10 | 83.32 | 27.42 | 49.95 | 59.81 | 41.03 | 65.31 | 74.28 |
| RN50x64 | softmax | 35.64 | 60.18 | 70.14 | 57.38 | 80.58 | 87.96 | 33.12 | 52.57 | 60.02 | 45.13 | 65.33 | 71.67 |
| | **MLOT(Ours)** | 43.06 | 70.26 | 79.56 | 57.98 | 81.12 | 88.06 | 41.64 | 65.49 | 74.67 | 54.03 | 77.35 | 84.61 |

## 4.2 CLIP-BASED ZERO-SHOT INFERENCE FOR TEXT-IMAGE RETRIEVAL

Image-Text Retrieval is a traditional multimodal task aimed at establishing an efficient correspondence between images and their descriptive text. Zero-shot retrieval aims to retrieve relevant items without any prior training on specific categories or datasets. Currently, this task has gained traction due to the increasing availability of pretrained models like CLIP (Radford et al., 2021).

In the traditional CLIP-based zero-shot retrieval process for Image-to-Text retrieval, the cosine similarity, multiplication of query image's embedding and all candidate captions' embedding, is used for predicting the most relevant match. However, this approach relies solely on two layers of information. To address this limitation, a new method for zero-shot retrieval is proposed. By augmenting the query image, we transform the Image-to-Text retrieval task into a Image-to-Text-to-Image MLOT problem. Additional layers of information are therefore incorporated. This multi-layered approach improves the retrieval recall by leveraging richer contextual information across multiple layers.

**Datasets and Experimental Setting.** For traditional downstream task image-text retrieval, we use COCO2017 (Lin et al., 2015) and Flickr30k (Young et al., 2014) dateset. For COCO2017, we use the 5k validation set, which has 5000 images and 25014 captions. For Flickr30, we use the whole 30k dataset, which has 31783 images and exactly 5 captions for each image. For our experiments, CLIP model (Radford et al., 2021) is employed to compute feature embedding of images and texts. Two different structure of CLIP: ViT-B/32 and RN50x64, are involved.

We propose a novel approach that leverages CLIP model to transform the retrieval problem into a Multi-Layered Optimal Transport (MLOT) problem. The cost metric in OT framework can be given by the negative cosine similarity between the normalized embeddings. We then can efficiently model the relationships between images and texts across multiple layers, as shown in Fig. 6. To formulate Multi-Layered OT problem, we implement data augmentation techniques. Specifically, we apply horizontal flipping to the images in the Image-to-Text task, and randomly select two captions from the available annotations for each image in the Text-to-Image task. The flipped query images or the synonymous cap-

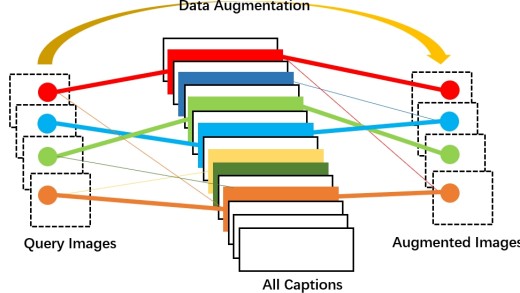

Figure 6: Procedures of converting Image-to-Text retrieval into MLOT problem.

tions are regarded as the third layer in MLOT. Therefore we effectively construct a $K = 3$ multi-layered transport scenario. Once the MLOT problem is formulated, we run the MLOT-Sinkhorn iterations. This MLOT problem ultimately returns two couplings, representing the similarity matches from the intermediate to the query and from the intermediate to the augmented query. A natural approach to handle this is to take their arithmetic mean as the final prediction, since this takes full advantage of the results from multiple layers.

**Baselines.** The CLIP model can be used for zero-shot image-text retrieval by computing cosine similarity in the embedding space. For baseline comparison, the softmax function is applied to the similarity scores to produce probability distributions, enabling the retrieval of the top-K image-caption pairs. The widely-used R@$k(k = 1, 5, 10)$ in cross-modal retrieval is reported for performance evaluation, which is the proportion of matched samples found in the top-K retrieved results.

**Experimental Results.** As shown in Tab. 4, the results demonstrate significant improvements in zero-shot retrieval compared to the softmax inference method across both COCO and Flickr30k datasets. Converting into MLOT problem significantly enhances the recall rate of zero-shot retrieval. On average, the recall rate is improved by 6.1% for the Transformer architecture and by 8.2% for the ResNet architecture across both retrieval tasks. These results indicate that our novel inference method, leveraging the MLOT framework, effectively captures the relationships between images and text.

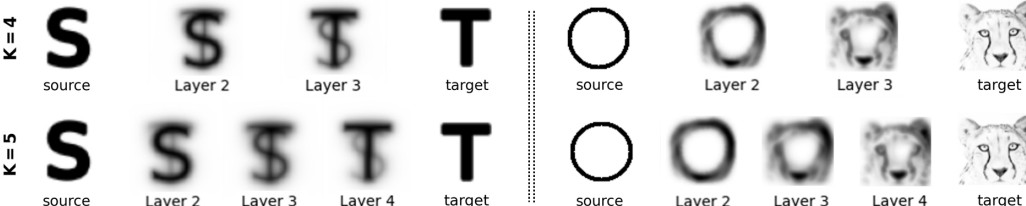

Figure 7: Intermediate images between given picture, generated by MLOT. The intermediate layers $(\mathbf{a}_k)_k]$) computed by MLOT-Sinkhorn are regarded as grayscale distribution of intermediate images. Top row: reformulating as 4 layers MLOT, which gives out 2 intermediate images. Second row: reformulating as 5 layers MLOT, which gives 3 intermediate images. Left: clean pictures transportation. Right: Transportation involved a more complex leopard image. Results demonstrate the effectiveness of reformulation as MLOT.

### 4.3 VISUAL EXPERIMENTS ON INTERMEDIATE DISTRIBUTIONS

Calculating image interpolation is a traditional task, aimed at generating intermediate images between two given input images, often used to create smooth transitions or fill in missing data. This task is mostly addressed by calculating the barycenter of two given images, where different weights are set to generate a coherent series of intermediate images. However. this method requires multiple computations with different weights. In contrast, we propose a new method based on MLOT, which can directly generate $K$ intermediate images in a single calculation. For a grayscale image, it can be viewed as a probability distribution vector of grayscale values. The gradual transition between two images exactly corresponds to the transport process of two grayscale value distributions. The transfer cost during this process should be determined by the relative distances between pixel locations. Thus, the cost matrix $D$ is defined on pixel-wise Euclidean distance between two 64x64 grid. Transitioning from a given image through several intermediate images to ultimately arrive at another image closely aligns with the MLOT problem. Therefore, a natural approach is to use $D$ as the cost matrix between any two layers, reformulating the task into solving the MLOT problem. Ultimately, the MLOT solution's intermediate layers should exactly represent the (grayscale distributions of the) intermediate images. We conducted tests on four grayscale images, each sized 64x64. As shown in Fig. 8, the left part features clean alphabet images, while the right part showcases a more complex leopard image. MLOT was applied varying 4 layers and 5 layers respectively, and the results indicate that our proposed method is effective. The intermediate layers calculated by MLOT can be directly interpreted as intermediate images at varying degrees of transition.

### 4.4 SUMMARY OF EXPERIMENTAL RESULTS

In this section, we summarize the key findings from our experiments. We confirmed the effectiveness and convergence of the MLOT-Sinkhorn algorithm, observing that it maintains a significant speed advantage while achieving a high level of accuracy (avg. error $< 0.7\%$). Furthermore, the algorithm can smoothly transition to precise solutions both for regularization on $\varepsilon$ and $\tau$. On the practical side, we highlighted the utility of the MLOT framework in two distinct tasks: Text-Image Retrieval and Intermediate Image Computing. By reformulating the original problems into a multi-layer structure, we significantly enhanced the utilization of intermediate information. In the zero-shot retrieval task, our approach achieved an average improvement of $7.2\%$ over Softmax. In the image-related task, we validated that the intermediate distributions in the MLOT solution visually represent interpolations between two images, providing an alternative method to compute interpolations without relying on Wasserstein barycenters.

## 5 CONCLUSION AND FUTURE WORK

In this paper, we propose Multi-layered Optimal Transport (MLOT), a novel approach extending traditional optimal transport to handle complex, multi-stage transportation scenarios. We then introduce the MLOT-Sinkhorn algorithm, leveraging entropic regularization for efficient computation on GPUs. Our method demonstrates superior performance in both speed and accuracy compared to existing solvers. Through experiments on zero-shot inference for Text-Image retrieval and intermediate image calculation, we validate MLOT's effectiveness and its potential to advance optimal transport applications in various fields. In future work, we believe that OT theory can be integrated and enhanced with a broader range of real-world scenarios, such as facility location problems (Cornuéjols et al., 1983), to enrich the application of matrix iteration algorithms based on OT in various operations research problems.

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

# A  GLOBAL CONVERGENCE OF MLOT-SINKHORN

This section study the convergence of entropic regularized OT.

## A.1  PROPERTY OF HILBERT METRIC

To measure the gap between iterative result and optimal coupling, Hilbert metric is introduced. $d_H(\mathbf{u}, \mathbf{u}') := \log \max_{i,j} \frac{\mathbf{u}_i \mathbf{u}'_j}{\mathbf{u}_j \mathbf{u}'_i}$. Firstly, several mathematical properties of Hilbert Metric are studied as follow.

1. $d_H\left(\frac{\mathbf{a}}{\mathbf{b}}, \frac{\mathbf{c}}{\mathbf{d}}\right) = d_H(\mathbf{ad}, \mathbf{bc}) \leqslant d_H(\mathbf{a}, \mathbf{c}) + d_H(\mathbf{b}, \mathbf{d})$
   **Proof**: By definition:
   $$LHS = \log \max \frac{\mathbf{a}_i \mathbf{c}_j \cdot \mathbf{b}_j \mathbf{d}_i}{\mathbf{b}_i \mathbf{d}_j \cdot \mathbf{a}_j \mathbf{c}_i} = d_H(\mathbf{ad}, \mathbf{cb})$$

   Separating the product, we have:
   $$LHS \leqslant \log \max \frac{\mathbf{a}_i \mathbf{c}_j}{\mathbf{a}_j \mathbf{c}_i} + \log \max \frac{\mathbf{b}_j \mathbf{d}_i}{\mathbf{b}_i \mathbf{d}_j} = d_H(\mathbf{a}, \mathbf{c}) + d_H(\mathbf{b}, \mathbf{d})$$

2. $d_H(\mathbf{a}^\varepsilon, \mathbf{b}^\varepsilon) = |\varepsilon| d_H(\mathbf{a}, \mathbf{b})$

   **Proof**: By definition: $LHS = \log \max \frac{\mathbf{a}_i^\varepsilon \mathbf{b}_j^\varepsilon}{\mathbf{a}_j^\varepsilon \mathbf{b}_i^\varepsilon}$. Since the operation is to maximize for all $i, j$, whether $\varepsilon > 0$ or $\varepsilon < 0$ will obtain the maximum or minimum at same row/column combination. Therefore the exponent can be separated out as absolute value.

3. $d_H(\mathbf{ta}, \mathbf{tb}) = d_H(t\mathbf{a}, t\mathbf{b})$

   **Proof**: If $t \in \mathbb{R}_+^n$ and $a, b \in \mathbb{R}_+^{n \times m}$. Then expand the by definition will prove this property straight forward. If $t \in \mathbb{R}_+^{w \times n}$, the situation becomes more complicated, which we will discuss immediately below.

## A.2 INTRODUCTION OF CONTRACTION RADIO

In the solution form $\mathrm{diag}(\mathbf{u}_k)\mathbf{S}_k\,\mathrm{diag}(\mathbf{v}_k)$, the constant argument $\mathbf{S}_k$ is critical in the convergence process. (Peyre & Cuturi, 2019) points out how matrix production influences Hilbert metric. (Franklin & Lorenz, 1989) generalizes this as a nature of a matrix, which can be regraded as contraction radio during iteration. As the following proposition shows.

$$d_H(\mathbf{S}\mathbf{v}, \mathbf{S}\mathbf{v}') \leq \lambda(\mathbf{S})d_H(\mathbf{v}, \mathbf{v}')$$

where $\lambda(\mathbf{S}) = \frac{\sqrt{\eta(\mathbf{S})}-1}{\sqrt{\eta(\mathbf{S})}+1}$ and $\eta(\mathbf{S}) := \max_{ijkl} \frac{\mathbf{S}_{ik}\mathbf{S}_{jl}}{\mathbf{S}_{jk}\mathbf{S}_{il}}$

The $\lambda(\mathbf{S})$ here is defined as

$$\sup\left\{\frac{d_H(\mathbf{S}\mathbf{y}, \mathbf{S}\mathbf{y}')}{d_H(\mathbf{y}, \mathbf{y}')}\ ,\ \mathbf{y}, \mathbf{y}' \in \mathbb{R}_+^n\right\}$$

, aiming to extract constant from Hilbert metric. Notice that $\lambda(\mathbf{S})$ is larger than 0 and less than 1, we call it contraction radio, denoted as $\gamma$.

## A.3 PROOF OF CONVERGENCE

**The case $\tau > 0$** Iteration steps (Suppose the $l$-th iteration):

$$\mathbf{u}_k^{l+1} = \mathbf{a}_k^l \oslash \mathbf{S}_k \mathbf{v}_k^l \tag{17}$$

$$\mathbf{v}_k^{l+1} = \mathbf{a}_k^l \oslash \mathbf{S}_k^\top \mathbf{u}_k^l \tag{18}$$

$$\mathbf{a}_k^{l+1} = \left(\mathbf{u}_k^{l+1} \odot \mathbf{v}_{k-1}^{l+1}\right)^{-\epsilon/\tau} \tag{19}$$

Denote the optimal value as $\mathbf{u}_k^*, \mathbf{v}_k^*, \mathbf{a}_k^*$. Now consider the Hilbert distance between $l+1$-th iteration to the optimal value:

$$d_H(\mathbf{u}^{l+1}, \mathbf{u}^*) = d_H\left(\frac{\mathbf{a}^l}{\mathbf{S}\mathbf{v}^l}, \frac{\mathbf{a}^*}{\mathbf{S}\mathbf{v}^*}\right) \tag{20}$$

$$\leqslant \lambda(\mathbf{S})\left[d_H\left(\mathbf{a}^l, \mathbf{a}^*\right) + d_H\left(\mathbf{v}^l, \mathbf{v}^*\right)\right] \tag{21}$$

$$d_H\left(\mathbf{v}^{l+1}, \mathbf{v}^*\right) = d_H\left(\frac{\mathbf{a}^l}{\mathbf{S}^\top \mathbf{u}^l}, \frac{\mathbf{a}^*}{\mathbf{S}^\top \mathbf{u}^*}\right) \tag{22}$$

$$\leqslant \lambda(\mathbf{S})\left[d_H\left(\mathbf{a}^l, \mathbf{a}^*\right) + d_H\left(\mathbf{u}^l \mathbf{u}^*\right)\right] \tag{23}$$

$$d_H\left(\mathbf{a}^l, \mathbf{a}^*\right) = d_H\left(\left(\mathbf{u}^l \odot \mathbf{v}^l\right)^{-\frac{\varepsilon}{\tau}}, (\mathbf{u}^* \odot \mathbf{v}^*)^{-\frac{\varepsilon}{\tau}}\right) \tag{24}$$

$$\leqslant \frac{\varepsilon}{\tau}\left[d_H\left(\mathbf{u}^l, \mathbf{u}^*\right) + d_H\left(\mathbf{v}^l, \mathbf{v}^*\right)\right] \tag{25}$$

The layer number $k$ is not important here, since we can simply replace all $\mathbf{a}_k^l, \mathbf{u}_k^l, \mathbf{v}_k^l, \gamma_k$ by the biggest one in this iteration, which guarantee a worst bound.

Substitute Eq. 23 into Eq. 25, we have:

$$d_H\left(\mathbf{a}^l, \mathbf{a}^*\right) \leqslant \frac{\varepsilon}{\tau}\frac{1+\gamma}{1-(\varepsilon/\tau)\gamma} \cdot d_H\left(\mathbf{u}^l, \mathbf{u}^*\right)$$

Substitute this into Eq. 21, finally we have:

$$d_H\left(\mathbf{u}^{l+1}, \mathbf{u}^*\right) \leqslant \frac{\gamma}{1-(\varepsilon/\tau)\gamma}\left(\gamma + \frac{2\varepsilon}{\tau}\gamma + \frac{\varepsilon}{\tau}\right) \cdot d_H\left(\mathbf{u}^l, \mathbf{u}^*\right)$$

Which indicates the Hilbert difference between $\mathbf{u}^l$ and optimal $\mathbf{u}^*$ converges in a exponential speed.

$$d_H\left(\mathbf{u}^{l+1}, \mathbf{u}^*\right) = \mathcal{O}\left[\left(\frac{\gamma}{1-(\varepsilon/\tau)\gamma}\left(\gamma + \frac{2\varepsilon}{\tau}\gamma + \frac{\varepsilon}{\tau}\right)\right)^l\right]$$

Since the contraction radio $\gamma$ is less than 1 (What's more, in experiment we find that $\gamma$ is always around $0.5\tilde{}0.7$), and $\varepsilon/\tau$ is always set less than 0.5, then $d_H\left(\mathbf{u}^{l+1}, \mathbf{u}^*\right) \to 0$.

**The case $\tau = 0$** Iteration steps (Suppose the $l$-th iteration):

$$\mathbf{u}_k^{l+1} = \mathbf{a}_k^l \oslash \mathbf{S}_k \mathbf{v}_k^l$$

$$\mathbf{v}_k^{l+1} = \mathbf{a}_k^l \oslash \mathbf{S}_k^\top \mathbf{u}_k^l \tag{26}$$

$$\mathbf{a}_k^{l+1} = \left((\mathbf{S}_{k-1}^\top \mathbf{u}_{k-1}^{l+1}) \odot (\mathbf{S}_k \mathbf{v}_k^{l+1})\right)^{1/2}$$

The remain proof is similar as the case $\tau > 0$.

$$
\begin{aligned}
d_H\left(\mathbf{a}^l, \mathbf{a}^*\right) &\leqslant \frac{1}{2}\gamma_{k-1}d_H\left(\mathbf{u}_{k-1}^{l+1}, \mathbf{u}_{k-1}^*\right) + \frac{1}{2}\gamma_k d_H\left(\mathbf{v}_k^{l+1}, \mathbf{v}_k^*\right) \\
&\leqslant \frac{1}{2}\gamma d_H\left(\mathbf{u}^{l+1}\right) + \frac{1}{2}\gamma d_H\left(\mathbf{v}^{l+1}\right)
\end{aligned}
\tag{27}
$$

, in which we denote $\max_k \gamma_k$ as $\gamma$, and represent all layer's Hilbert distance by the biggest one in this iteration $d_H\left(\mathbf{a}^l, \mathbf{a}^*\right)$, etc. We have:

$$
(2 - 2\gamma^2 - \gamma^3)d_H\left(\mathbf{a}^l, \mathbf{a}^*\right) \leqslant \gamma^2(1+\gamma)d_H\left(\mathbf{u}^l, \mathbf{u}^*\right)
\tag{28}
$$

Combine Eq. 21, Eq. 23 and Eq. 28, finally we have:

$$
d_H\left(\mathbf{u}^{l+1}, \mathbf{u}^*\right) \leqslant \frac{\gamma^2(\gamma+2)}{2 - 2\gamma^2 - \gamma^3} \cdot d_H\left(\mathbf{u}^l, \mathbf{u}^*\right)
$$

Which indicates the Hilbert distance between $\mathbf{u}^l$ and optimal $\mathbf{u}^*$ converges in a exponential speed.

$$
d_H\left(\mathbf{u}^{l+1}, \mathbf{u}^*\right) = \mathcal{O}\left[\left(\frac{\gamma^2(\gamma+2)}{2 - 2\gamma^2 - \gamma^3}\right)^l\right]
$$

## B    PROOF OF PROP.1

**The case $\tau = 0$.**

The entropic regularized MLOT can be formulated as

$$
\min_{\{\mathbf{P}_k\},\{\mathbf{a}_k\}} \sum_{k=1}^{K-1}\left( <\mathbf{C}_k, \mathbf{P}_k> -\epsilon H(\mathbf{P}_k)\right) - \tau \sum_{k=2}^{K-1} H(\mathbf{a}_k)
\tag{29}
$$

subject to

$$
\mathbf{P}_k\mathbf{1} = \mathbf{a}_k \quad \text{and} \quad \mathbf{P}_k^\top\mathbf{1} = \mathbf{a}_{k+1} \quad \forall k = 1,\ldots,K-1.
\tag{30}
$$

The Lagrange multiplier function is

$$
\begin{aligned}
L = &\sum_{k=1}^{K-1}\left( <\mathbf{C}_k, \mathbf{P}_k> -\epsilon H(\mathbf{P}_k)\right) - \tau \sum_{k=2}^{K-1} H(\mathbf{a}_k) \\
&- \sum_{k=1}^{K-1} <\mathbf{f}_k, \mathbf{P}_k\mathbf{1} - \mathbf{a}_k> - <\mathbf{g}_k, \mathbf{P}_k^\top\mathbf{1} - \mathbf{a}_{k+1}>
\end{aligned}
\tag{31}
$$

Firstly,

$$
\begin{aligned}
\frac{\partial L}{\partial \mathbf{P}_k} &= \mathbf{C}_k + \varepsilon\log\mathbf{P}_k - \mathbf{f}_k\mathbf{1}^\top - \mathbf{1}^\top\mathbf{g}_k = 0 \\
&\Rightarrow \mathbf{P}_k = \text{Diag}\left(e^{\mathbf{f}_k/\varepsilon}\right) \cdot e^{-\mathbf{C}_k/\varepsilon} \cdot \text{Diag}\left(e^{\mathbf{g}_k/\varepsilon}\right)
\end{aligned}
\tag{32}
$$

Set that: $\mathbf{u}_k = e^{\mathbf{f}_k/\varepsilon}, \mathbf{v}_k = e^{\mathbf{g}_k/\varepsilon}, \mathbf{S}_k = e^{-\mathbf{C}_k/\varepsilon}$, we have:

$$
\mathbf{P}_k = \text{Diag}(\mathbf{u}_k)\mathbf{S}_k\text{Diag}(\mathbf{v}_k)
\tag{33}
$$

Due to $\mathbf{P}_k\mathbf{1} = \mathbf{a}_k$ and $\mathbf{P}_k^\top\mathbf{1} = \mathbf{a}_{k+1}$ We have:

$$
\mathbf{u}_k = \frac{\mathbf{a}_k}{\mathbf{S}_k\mathbf{v}_k}, \quad \mathbf{u}_k = \frac{\mathbf{a}_{k+1}}{\mathbf{S}_k^\top\mathbf{u}_k}
\tag{34}
$$

What's more, when $\tau = 0$:

$$
\frac{\partial L}{\partial \mathbf{a}_k} = \mathbf{f}_k + \mathbf{g}_{k-1} = 0
\tag{35}
$$

Thus, $\mathbf{u}_k \odot \mathbf{v}_{k-1} = \mathbf{1}$ Then we have:

$$
\frac{\mathbf{a}_k}{\mathbf{S}_k\mathbf{v}_k} \odot \frac{\mathbf{a}_k}{\mathbf{S}_{k-1}^\top\mathbf{u}_{k-1}} = 1
$$

$$
\mathbf{a}_k = \left[\frac{\mathbf{a}_k}{\mathbf{S}_k\mathbf{v}_k} \odot \frac{\mathbf{a}_k}{\mathbf{S}_{k-1}^\top\mathbf{u}_{k-1}}\right]^{\frac{1}{2}}, \quad \text{for} k = 2, ..., K-1
\tag{36}
$$

**The case $\tau > 0$.**

The Lagrange multiplier function is

$$
L = \sum_{k=1}^{K-1} \Big( <\mathbf{C}_k, \mathbf{P}_k> -\epsilon H(\mathbf{P}_k) \Big) - \tau \sum_{k=2}^{K-1} H(\mathbf{a}_k)
$$
$$
- \sum_{k=1}^{K-1} <\mathbf{f}_k, \mathbf{P}_k \mathbf{1} - \mathbf{a}_k> - <\mathbf{g}_k, \mathbf{P}_k^\top \mathbf{1} - \mathbf{a}_{k+1}> \tag{37}
$$

Firstly,

$$
\frac{\partial L}{\partial \mathbf{P}_k} = \mathbf{C}_k + \varepsilon \log \mathbf{P}_k - \mathbf{f}_k \mathbf{1}^\top - \mathbf{1}^\top \mathbf{g}_k = 0
$$
$$
\Rightarrow \mathbf{P}_k = \mathrm{Diag}\left(e^{\mathbf{f}_k/\varepsilon}\right) \cdot e^{-\mathbf{C}_k/\varepsilon} \cdot \mathrm{Diag}\left(e^{\mathbf{g}_k/\varepsilon}\right) \tag{38}
$$

Set that: $\mathbf{u}_k = e^{\mathbf{f}_k/\varepsilon}, \mathbf{v}_k = e^{\mathbf{g}_k/\varepsilon}, \mathbf{S}_k = e^{-\mathbf{C}_k/\varepsilon}$, we have:

$$
\mathbf{P}_k = \mathrm{Diag}(\mathbf{u}_k) \mathbf{S}_k \mathrm{Diag}(\mathbf{v}_k) \tag{39}
$$

Due to $\mathbf{P}_k \mathbf{1} = \mathbf{a}_k$ and $\mathbf{P}_k^\top \mathbf{1} = \mathbf{a}_{k+1}$ We have:

$$
\mathbf{u}_k = \frac{\mathbf{a}_k}{\mathbf{S}_k \mathbf{v}_k}, \quad \mathbf{u}_k = \frac{\mathbf{a}_{k+1}}{\mathbf{S}_k^\top \mathbf{u}_k} \tag{40}
$$

What's more, when $\tau > 0$

$$
\frac{\partial L}{\partial \mathbf{a}_k} = \tau \log \mathbf{a}_k + \mathbf{f}_k + \mathbf{g}_{k-1} = 0
$$
$$
\mathbf{a}_k = (\mathbf{u}_k \odot \mathbf{v}_{k-1})^{-\epsilon/\tau} \tag{41}
$$

## C  PROOF OF PROP.2

From the definition of $\widetilde{KL}$ and $(\mathbf{S}_k)_{ij} = e^{-(\mathbf{C}_k)_{ij}/\epsilon}$, we have

$$
\sum_{k=1}^{K-1} \widetilde{KL}(\mathbf{P}_k | \mathbf{S}_k) = \sum_{k=1}^{K-1} \sum_{ij} \left( (\mathbf{P}_k)_{ij} \log(\mathbf{P}_k)_{ij} - (\mathbf{P}_k)_{ij} + (\mathbf{P}_k)_{ij} \frac{(\mathbf{C}_k)_{ij}}{\varepsilon} + (\mathbf{S}_k)_{ij} \right)
$$
$$
= \sum_{k=1}^{K-1} \sum_{ij} \left( (\mathbf{P}_k)_{ij} (\log(\mathbf{P}_k)_{ij} - 1) + \frac{1}{\epsilon} (\mathbf{P}_k)_{ij} (\mathbf{C}_k)_{ij} + (\mathbf{S}_k)_{ij} \right) \tag{42}
$$
$$
= \frac{1}{\epsilon} \sum_{k=1}^{K-1} <\mathbf{C}_k, \mathbf{P}_k> -\varepsilon H(\mathbf{P}_k) + \mathrm{Const} .
$$

and

$$
\sum_{k=2}^{K-1} \widetilde{KL}(\mathbf{a}_k | \boldsymbol{\Delta}_k) = \sum_{k=2}^{K-1} \sum_i \mathbf{a}_k)_i (\log \mathbf{a}_k)_i + \log n_k - 1)
$$
$$
= \sum_{k=2}^{K-1} \sum_i \mathbf{a}_k)_i (\log \mathbf{a}_k)_i - 1) + \log n_k \sum_i \mathbf{a}_k)_i \tag{43}
$$
$$
= \frac{1}{\tau} \sum_{k=2}^{K-1} H(\mathbf{a}_k) + \mathrm{Const} .
$$

Notice that the Const in expression is irrelevant when it comes to solving optimization problems. Therefore $\min_{(\mathbf{P}_k)_k, (\mathbf{a}_k)_k} \varepsilon \sum_{k=1}^{K-1} \widetilde{KL}(\mathbf{P}_k | \mathbf{S}_k) + \tau \sum_{k=2}^{K-1} \widetilde{KL}(\mathbf{a}_k | \boldsymbol{\Delta}_k)$ is exactly equivalent to Eq. 7.

## D  PROOF OF PROP.3

**Convergence with $\varepsilon$**  In this part, we prove that the entropic regularization on couplings will converge to original MLOT. We consider a sequence $(\varepsilon_l)_l > 0$ such that $\varepsilon_l \to 0$. We denote $(\mathbf{P}_k^{\varepsilon_l})_k$ as the optimal solution

of Eq. 7 with $\varepsilon = \varepsilon_l, \tau = 0$, and denote $(\mathbf{P}_k^\star)_k$ as the optimal solution of Eq. 5. By optimality of $(\mathbf{P}_k^{\varepsilon_l})_k$ and $(\mathbf{P}_k^\star)_k$ for their respective optimization problems, we have:

$$\sum_{k=1}^{K-1} < \mathbf{C}_k, \mathbf{P}_k^{\varepsilon_l} > -\varepsilon_l H(\mathbf{P}_k^{\varepsilon_l}) \quad \leqslant \quad \sum_{k=1}^{K-1} < \mathbf{C}_k, \mathbf{P}_k^\star > -\varepsilon_l H(\mathbf{P}_{k\star})$$

$$\sum_{k=1}^{K-1} < \mathbf{C}_k, \mathbf{P}_k^\star > \quad \leqslant \quad \sum_{k=1}^{K-1} < \mathbf{C}_k, \mathbf{P}_k^{\varepsilon_l} > \tag{44}$$

Therefore:

$$0 \leqslant \sum_{k=1}^{K-1} < \mathbf{C}_k, \mathbf{P}_k^{\varepsilon_l} - \mathbf{P}_k^\star > \quad \leqslant \quad \sum_{k=1}^{K-1} \varepsilon_l \left[ H(\mathbf{P}_k^{\varepsilon_l}) - H(\mathbf{P}_k^\star) \right] \tag{45}$$

Since entropic function $H(\mathbf{P})$ is continuous and inner product here is always positive, the limitation $\varepsilon_l \to 0$ shows that $\mathbf{P}_k^{\varepsilon_l} = \mathbf{P}_k^\star, \quad \forall k = 1, 2, ..., K-1$, which proves Eq. 12.

**Convergence with $\tau$**  In this part, we prove that the entropic regularization on both couplings and intermediates will converge to the problem that only regularize couplings, given the fixed $\varepsilon_0$. We consider a sequence $(\tau_l)_l > 0$ such that $\tau_l \to 0$. We denote $(\mathbf{P}_k^{\tau_l})_k$ as the optimal solution of Eq. 7 with $\varepsilon = \varepsilon_0, \tau = \tau_l$, and denote $(\mathbf{P}_k^{\varepsilon_0})_k$ as the optimal solution of Eq. 7 without regularization on intermediates. By optimality of $(\mathbf{P}_k^{\tau_l})_k$ and $(\mathbf{P}_k^{\varepsilon_0})_k$ for their respective optimization problems, we have:

$$\sum_{k=1}^{K-1} < \mathbf{C}_k, \mathbf{P}_k^{\tau_l} > -\varepsilon_0 H(\mathbf{P}_k^{\tau_l}) - \tau_l \sum_{k=2}^{K-1} H(\mathbf{a}_k^{\tau_l}) \quad \leqslant \quad \sum_{k=1}^{K-1} < \mathbf{C}_k, \mathbf{P}_k^{\varepsilon_0} > -\varepsilon_0 H(\mathbf{P}_k^{\varepsilon_0}) - \tau_l \sum_{k=2}^{K-1} H(\mathbf{a}_k^{\varepsilon_0})$$

$$\sum_{k=1}^{K-1} < \mathbf{C}_k, \mathbf{P}_k^{\varepsilon_0} > -\varepsilon_0 H(\mathbf{P}_k^{\varepsilon_0}) \quad \leqslant \quad \sum_{k=1}^{K-1} < \mathbf{C}_k, \mathbf{P}_k^{\tau_l} > -\varepsilon_0 H(\mathbf{P}_k^{\tau_l}) \tag{46}$$

Therefore:

$$0 \leqslant \sum_{k=1}^{K-1} < \mathbf{C}_k, \mathbf{P}_k^{\tau_l} - \mathbf{P}_k^{\varepsilon_0} > -\varepsilon_0 \left[ H(\mathbf{P}_k^{\tau_l}) - H(\mathbf{P}_k^{\varepsilon_0}) \right] \quad \leqslant \quad \sum_{k=2}^{K-1} \tau_l \left[ H(\mathbf{a}_k^{\tau_l}) - H(\mathbf{a}_k^{\varepsilon_0}) \right] \tag{47}$$

Similarly, since entropic function $H(\mathbf{a})$ is continuous, the limitation $\tau_l \to 0$ shows that regularization on intermediate can converge to non-regularization on intermediate:

$$\sum_{k=1}^{K-1} < \mathbf{C}_k, \mathbf{P}_k^{\tau_l} > -\varepsilon_0 H(\mathbf{a}_k^{\tau_l}) = \sum_{k=1}^{K-1} < \mathbf{C}_k, \mathbf{P}_k^{\varepsilon_0} > -H(\mathbf{a}_k^{\varepsilon_0}).$$

# E  ARCHIMEDEAN DISTANCE BETWEEN TWO POINTS

Archimedes' spiral is curve expressed as $r(\theta) = b(\theta - \theta_0)$. Suppose two a spiral passes through two points $(r_1, \theta_1)$, $(r_2, \theta_2)$. The curve's parameters can be determined as:

$$b = \frac{r_2 - r_1}{\theta_2 - \theta_1}, \quad \theta_0 = \frac{\theta_1 r_2 - \theta_2 r_1}{r_2 - r_1} \tag{48}$$

The length of the curve is:

$$\mathrm{d}l = \sqrt{\mathrm{d}r^2 + (r\mathrm{d}\theta)^2}$$

$$\Rightarrow \quad L = \int_{r_1}^{r_2} \sqrt{1 + \frac{r^2}{b^2}} \mathrm{d}r \tag{49}$$

$$= \frac{r}{2b} \sqrt{b^2 + r^2} + \frac{b}{2} \ln \left( r + \sqrt{b^2 + r^2} \right) \Bigg|_{r_1}^{r_2}$$

Under the circumstances in Ring Data, where the radii of neighbouring rings differ by 1, thus $b = 1/(\theta_2 - \theta_1)$. Further denote $\theta_2 - \theta_1$ as $a$. Let:

$$F(r) = \frac{r}{2} \sqrt{1 + a^2 r^2} + \frac{1}{2a} \ln \left( ar + \sqrt{1 + a^2 r^2} \right) \tag{50}$$

Then the Archimedean distance between two points can be written as $F(r_2) - F(r_1)$.

# F  SUPPLEMENTARY EXPERIMENTS FOR REBUTTAL

Table 3: Solving the reformulated MLOT problem returns 2 coupling $\mathbf{P}_1, \mathbf{P}_2$, which are regarded as probability prediction metrix. In our submission, $\mathbf{P}_1 + \mathbf{P}_2^\top$ is adopted for final prediction. This table displays the $R@k$ results using single coupling (either $\mathbf{P}_1$ or $\mathbf{P}_2$) instead. And the table below shows how many predicted labels are same among $\mathbf{P}_1$ and $\mathbf{P}_2$.

| | | COCO | | | | | | Flickr30k | | | | | |
| | | Text⇒Image | | | Image⇒Text | | | Text⇒Image | | | Image⇒Text | | |
| Structure | Inference | R@1 | R@5 | R@10 | R@1 | R@5 | R@10 | R@1 | R@5 | R@10 | R@1 | R@5 | R@10 |
|---|---|---|---|---|---|---|---|---|---|---|---|---|---|
| ViT-B/32 | softmax | 29.02 | 52.84 | 64.26 | 49.82 | 74.64 | 83.10 | 24.42 | 42.96 | 51.00 | 34.34 | 54.44 | 61.97 |
| | MLOT(Ours) | 35.10 | 61.22 | 72.18 | 50.66 | 75.10 | 83.32 | 27.42 | 49.95 | 59.81 | 41.03 | 65.31 | 74.28 |
| | 1-Coupling Pred | 29.0\|28.6 | 52.4\|52.1 | 64.3\|62.8 | 49.7\|49.9 | 74.7\|74.6 | 83.3\|83.3 | 21.5\|21.7 | 41.0\|41.1 | 50.6\|50.6 | 40.9\|37.6 | 64.8\|61.2 | 73.8\|70.7 |
| RN50x64 | softmax | 35.64 | 60.18 | 70.14 | 57.38 | 80.58 | 87.96 | 33.12 | 52.57 | 60.02 | 45.13 | 65.33 | 71.67 |
| | MLOT(Ours) | 43.06 | 70.26 | 79.56 | 57.98 | 81.12 | 88.06 | 41.64 | 65.49 | 74.67 | 54.03 | 77.35 | 84.61 |
| | 1-Coupling Pred | 35.8\|35.8 | 60.7\|60.1 | 71.1\|70.6 | 58.0\|56.6 | 81.1\|80.4 | 88.3\|86.9 | 33.7\|33.5 | 55.8\|55.8 | 65.0\|65.3 | 53.7\|50.9 | 77.1\|74.7 | 84.5\|82.4 |

| | COCO | | Flckr30k | |
| Backbone Structure | Text⇒Image | Image⇒Text | Text⇒Image | Image⇒Text |
|---|---|---|---|---|
| ViT-B/32 | 38.6 (19319/5000 × 10) | 80.2 (40083/5000 × 10) | 38.4 (122114/10 × 31783) | 54.5 (171811/10 × 31783) |
| RN50x64 | 39.7 (19851/5000 × 10) | 78.6 (39299/5000 × 10) | 39.8 (126438/10 × 31783) | 65.8 (209178/10 × 31783) |

Table 4: The wall-clock computation time for the Image-Text Retrieval task

| | | COCO | | Flickr30k | |
| Structure | Inference | Text⇒Image | Image⇒Text | Text⇒Image | Image⇒Text |
|---|---|---|---|---|---|
| ViT-B/32 | softmax | 5.12 | 68.50 | 57.12 | 33.15 |
| | MLOT(Ours) | 19.32 | 439.37 | 229.31 | 1128.97 |
| RN50x64 | softmax | 7.81 | 205.55 | 52.65 | 28.09 |
| | MLOT(Ours) | 46.06 | 1131.11 | 262.74 | 2175.40 |

Table 5: Experiment on synthetic Line dataset, with different setting of layers shape (Differs from experiment in the paper, whose source/target node's number is set to be smallest).

| | | Gurobi | | MLOT($\tau = 0$) | | MLOT($\tau > 0$) | |
| N | $(n_k)_k$ | Objective | Time(s) | Objective | Time(s) | Objective | Time(s) |
|---|---|---|---|---|---|---|---|
| 1500 | $[500, 250, 250, 500]$ | 0.8694 | 7.163 | 0.8733 | 4.091 | 0.8746 | 3.279 |
| 2600 | $[800, 500, 500, 800]$ | 0.7705 | 25.329 | 0.7753 | 3.268 | 0.7747 | 4.034 |
| 3000 | $[1000, 500, 500, 1000]$ | 0.4159 | 29.312 | 0.4261 | 8.137 | 0.4238 | 10.059 |
| 3600 | $[1000, 800, 800, 1000]$ | 0.4087 | 59.697 | 0.4213 | 8.466 | 0.4191 | 10.331 |
| 4000 | $[1000, 1000, 1000, 1000]$ | 0.4161 | 163.600 | 0.4257 | 8.520 | 0.4247 | 10.282 |
| 5000 | $[1500, 1000, 1000, 1500]$ | 0.1594 | 148.347 | 0.1719 | 8.278 | 0.1690 | 10.422 |
| 6000 | $[2000, 1000, 1000, 2000]$ | 0.0950 | 167.729 | 0.1124 | 9.955 | 0.1086 | 14.366 |

Figure 8: Comparison of two methods computing intermediate images: via Barycenter and via MLOT. The figure shows the situation of $K = 5$ (need to generate 3 images). The top row shows the results generated by MLOT, where **all intermediates images are computed within single computation procedure.** The botton row shows the results generated by Barycenter, where 3 times of computation is needed, setting $\lambda = 0.25, 0.5, 0.75$ respectively.

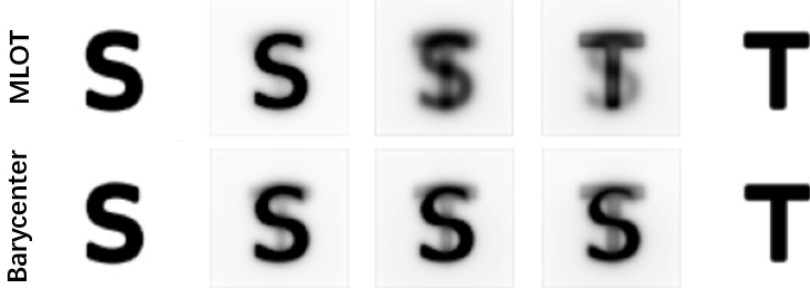

Table 6: Experiment on synthetic Line dataset with layer number $K = 3$. We randomly generate flow constraints $\mathbf{s}_k$ for each layer, i.e. $\mathbf{a}_k \leqslant \mathbf{s}_k,\ k = 2, 3 \ldots K - 1$. Our proposed MLOT-Sinkhorn still provide a highly accurate result compared with Gurobi. This demonstrate **our algorithm's adaptability towards constraints on flow**.

| | **Without--Constraints** | **Adding Random Constraints** | | | |
| | | **MLOT$(\tau = 0)$** | | **Gurobi** | |
| N | Objective | Objective | Time(s) | Objective | Time(s) |
|---|---|---|---|---|---|
| $1 \times 10^3$ | 0.6447 | 0.9349 | 12.3 | 0.9331 | 5.5 |
| $2 \times 10^3$ | 0.6305 | 0.9458 | 16.4 | 0.9450 | 26.5 |
| $3 \times 10^3$ | 0.3426 | 0.8426 | 19.8 | 0.8395 | 68.9 |
| $4 \times 10^3$ | 0.3667 | 0.7216 | 20.6 | 0.7188 | 193.8 |
| $5 \times 10^3$ | 0.1550 | 0.7306 | 18.4 | 0.7206 | 417.5 |
| $6 \times 10^3$ | 0.5088 | 0.5456 | 31.2 | 0.5426 | 818.5 |

## G  STATIC SCHRÖDINGER BRIDGE PROBLEM AND MLOT

The SB problem is a classical problem. In the discrete-time setting, given density

$$p(x_{0:N}) = p_0(x_0) \prod_{k=0}^{N-1} p_{k+1|k}(x_{k+1} \mid x_k)$$

which describes the process adding noise to the data. We aim to find $\pi^\star \in P_{N+1}$ such that:

$$\pi^\star = \arg\min \{ \mathrm{KL}(\pi \mid p) : \pi \in P_{N+1}, \pi_0 = p_{\text{data}}, \pi_N = p_{\text{prior}} \}$$

This dynamic formulation admits a static analogue:

$$\pi^{s,\star} = \arg\min \{ \mathrm{KL}(\pi^s \mid p_{0,N}) : \pi^s \in P_2, \pi_0^s = p_{\text{data}}, \pi_N^s = p_{\text{prior}} \}$$

Solving the full Schrodinger Bridge problem, especially in its continuous form, can be computationally difficult. Several numerical methods were proposed, such as IPF (Chen et al., 2021), DifussionSB (De Bortoli et al., 2023).

Under mild assumptions, the static SB problem can be seen as an entropy-regularized optimal transport problem:

$$\pi^{\mathrm{s},\star} = \arg\min \left\{ -\mathbb{E}_{\pi^\star}\left[\log p_{N|0}\left(X_N \mid X_0\right)\right] - \mathrm{H}\left(\pi^{\mathrm{s}}\right) : \pi^{\mathrm{s}} \in P_2, \pi_0^{\mathrm{s}} = p_{\mathrm{data}}, \pi_N^{\mathrm{s}} = p_{\mathrm{prior}} \right\}$$

The KL form of MLOT problem is presented in Eq. 2:

$$\min_{(\mathbf{P}_k)_k,(\mathbf{a}_k)_k} \varepsilon \sum_{k=1}^{K-1} \widetilde{KL}(\mathbf{P}_k|\mathbf{S}_k) + \tau \sum_{k=2}^{K-1} KL(\mathbf{a}_k|\mathbf{\Delta}_k)$$

As proved in Appendix.C, minimizing the part $\tau KL(\mathbf{a}_k|\Delta_k)$ is equivalent to minimizing $H(\mathbf{a}_k)$. Therefore, our MLOT problem shares a **similar KL divergence structure to the discrete form of the Schrödinger bridge.**

By drawing this parallel, we suggest that in the special case where SB problem is discrete, MLOT-Sinkhorn provides a potential approach to solving the SB problem.

