# OpenReview forum: "MLOT: Extending the Bipartite Structure towards Multi-Layered Structure for Optimal Transport"
_ICLR.cc/2025/Conference — Submitted to ICLR 2025_

### Official Review · Reviewer_b2Pr · 2024-10-28

**Soundness:** 3
**Presentation:** 3
**Contribution:** 2
**Rating:** 6
**Confidence:** 3

**Summary:**

This paper introduces a Multi-Layered Optimal Transport (MLOT) problem and proposes a Sinkhorn algorithm for solving this problem. The MLOT problem generalizes the standard OT problem by modeling transport from the source distribution to the target distribution through the multiple intermediate layers. The proposed Sinkhorn algorithm offers superior efficiency on complex problems while maintaining comparable accuracy to the LP solver, such as Gurobi. The proposed method is also evaluated on real-world tasks, such as Text-Image retrieval and intermediate image computation.

**Strengths:**

- This paper introduces an interesting variant of the standard OT problem and demonstrates its practical utility by applying the MLOT framework to real-world tasks.
- This paper presents theoretical results on the convergence of the proposed algorithm.
- This paper is easy to follow.

**Weaknesses:**

- Please refer to the questions section.

**Questions:**

- Could you provide the wall-clock computation time for the Image-Text Retrieval task in Table 2? The proposed method is likely to be slower compared to the standard softmax operation.
- In the Image-Text Retrieval task (Fig 6), the two couplings (the coupling between Query image and Caption, and the coupling between Caption and Augmented image) can indicate different image-caption pairings. How coherent are these two couplings?
- In the Image-Text Retrieval task, the standard OT problem can be directly applied between the Query image and Caption using the same cost as in the MLOT, i.e., the negative cosine similarity between the normalized embeddings. Is there a meaningful difference in performance between the MLOT and OT frameworks (Table 2)?
- In the intermediate image experiments (Fig 7), how do the MLOT results differ from the standard OT barycenter?
- In the synthetic data experiments (Sec 4.1), the data points in the intermediate layers are also sampled, e.g., $(n\_{k})\_{k} = \\{3, 10, 20, 20, 10, 3\\}$ in Line 373. Could you provide the reason for sampling these intermediate samples? Is it just for setting the support of the intermediate distributions $a\_{k}$?

---

> ### Author Response · Authors · 2024-11-27
> **Part 1 of Response to Reviewer b2Pr**
>
> We sincerely appreciate your time and thorough review. We hope the following answers help clarify the points you raised.
>
> >***Q1: Could you provide the wall-clock computation time for the Image-Text Retrieval task in Table 2? The proposed method is likely to be slower compared to the standard softmax operation.***
>
> A1: Thank you for the question. We conducted additional experiments to measure the wall-clock computation time for the Image-Text Retrieval task, and the results are provided in ***Appendix. F Table 4***.
>
> You are correct that formulating the retrieval task as an MLOT problem increases the inference time compared to directly using the standard softmax operation. However, it is important to emphasize that our proposed method is **training-free**, which only modifies the inference process and does not require any additional training to improve retrieval accuracy.
>
> In comparison to other methods that also employ data augmentation to enhance CLIP’s performance, such as MixGen$^1$, CLIP-MedFake$^2$ etc., which require additional training, our algorithm does not involve any retraining of the model. We believe the extra time required by our algorithm in inference is a reasonable trade-off given the observed improvements in R@k.
>
> >***Q2: In the Image-Text Retrieval task, the two couplings can indicate different image-caption pairings. How coherent are these two couplings?***
>
> A2: As shown in ***Appendix. F Table 3***, we record the respective prediction results from 2 MLOT coulings (Since 3 layers in total) separately.
>
> As shown in the darker rows of the first table, **using individual OT predictions alone does not significantly improve accuracy; only when both are used in MLOT do we observe a noticeable improvement.**
>
> The second table shows how many predictions are identical among the 10 preidictions given by two couplings respectively. For Text$\Rightarrow$Image task, the overlap between the two coupling is relatively low, thus combining them (by averaging) leads to a more significant performance improvement. For Image$\Rightarrow$Text task, they are more coherent, and averaging them yields a less improvement.
>
> >***Q3:In the Image-Text Retrieval task, the standard OT problem can be directly applied between the Query image and Caption using the same cost as in the MLOT, i.e., the negative cosine similarity between the normalized embeddings. Is there a meaningful difference in performance between the MLOT and OT frameworks (Table 2)?***
>
> A3: Thank you for your question. As shown in the above experiment tables, the darker rows in the first table indicate that using a single OT coupling prediction does not outperform the softmax-based approach. It is only when we leverage all the coupling information from MLOT, a significant performance improvement is achieved. This highlights the added benefit of considering multiple couplings in the MLOT framework.
>
> ---
> 1. https://openaccess.thecvf.com/content/WACV2023W/Pretrain/papers/Hao_MixGen_A_New_Multi-Modal_Data_Augmentation_WACVW_2023_paper.pdf
> 2. H. Chen et al., "Clip-Medfake: Synthetic Data Augmentation With AI-Generated Content for Improved Medical Image Classification," 2024 IEEE International Conference on Image Processing

---

> > ### Author Response · Authors · 2024-11-27
> > **Part 2 of Response to Reviewer b2Pr**
> >
> > >***Q4:In the intermediate image experiments (Fig 7), how do the MLOT results differ from the standard OT barycenter?***
> >
> > A4: Thank you for your comment. We present experimental results comparing the traditional approach of computing intermediate images via **barycenter** with our method using **MLOT**. As shown in ***Appendix. F Figure 8***, MLOT successfully performs this task and achieves slight improvements in smoothness.
> >
> > However, the main contribution of our work is that, **MLOT allows for the simultaneous computation of all intermediate images in a single step**. In traditional methods via barycenter, it requires repeated computations for each intermediate image with varying $\lambda$ (For each image, we need set cost metric as $[\lambda D,(1-\lambda)D]$). In contrast, MLOT can solve all intermediates by setting cost metric as $[D,D\dots,D]$
> >
> > |  K  | Barycenter(Second) | MLOT(Second) |
> > |:---:|:------------------:|:------------:|
> > |  3  |       12.81        |    12.79     |
> > |  4  |       25.59        |    19.36     |
> > |  5  |       38.67        |    25.26     |
> > |  6  |       51.12        |    31.60     |
> > |  8  |       77.15        |    43.24     |
> > | 10  |       102.21       |    53.83     |
> >
> >
> > The table above shows the time differences between the two methods for generating varying numbers of intermediate images. When $K=3$ (only 1 intermediate image needs to be generated), both methods demonstrate equivalent time complexity. However, when $K>3$, which requires generating more than one intermediate image, **the time advantage of MLOT becomes significant.**
> >
> > >***Q5: In the synthetic data experiments (Sec 4.1), the data points in the intermediate layers are also sampled, e.g., $(n_k)_k = \{3,10,20,20,10,3\}$ in Line 373. Could you provide the reason for sampling these intermediate samples?***
> >
> > A5: The setting of $(n_k)_k$ can theoretically be arbitrary. as our method is not constrained by the shape of the intermediate layers. To demonstrate this, we conducted additional experiments, shown in ***Appendix. F Table 5***. The new experiment was conducted under circumstances where source/target's node number is larger than intermediates', and the accuracy and time cost is recorded. The results demonstrate that, our algorithm's accuracy and time cost is not affected by changing $n_k$ setting.
> >
> > The specific choice of $(n_k)_k = \{3,10,20,20,10,3\}$ in Line 373 was mainly trying to make ***Fig. 2*** visually clear. Since we set the flow at each source/target's node to 1, if thier number is larger than the number of intermediate nodes, it's not clear to show the assignments and transportation choices.

---

> > > ### Comment · Reviewer_b2Pr · 2024-11-30
> > >
> > > I appreciate the authors for the clarifications and additional experiments. These have been helpful in addressing my concerns.

---

### Official Review · Reviewer_Prsw · 2024-11-01

**Soundness:** 2
**Presentation:** 2
**Contribution:** 3
**Rating:** 5
**Confidence:** 3

**Summary:**

The paper introduces multi-layered traditional optimal transport, which finds an optimal transport map between the input and output distribution through multiple layers as an extension of traditional optimal transport. Compared with other optimal transport methods involving multiple distributions, the authors introduce uncertain intermediate distributions computed based on the previous and the next distributions. Additionally, the authors further propose MLOT-Sinkorn, which is designed specifically for MLOT to accelerate computation. The experimental results show that MLOT achieved better results than previous optimization methods.

**Strengths:**

1. This paper introduces a variant of multiple distributions optimal transport and a corresponding variant of the Sinkhorn algorithm to solve this problem.
2. The results of the method significantly outperform existing methods for multiple distributions.
3. The experimental section is thorough and clearly written, detailing the settings and configurations for each task.

**Weaknesses:**

1. The motivation for the introduction of MLOT compared with other multiple distributions for optimal transport is not written. Please refer to Question 1 in the Questions section for more details.
2. The related works section doesn't mention any previous works on multiple distributions in optimal transport although in the abstract, the authors state that "Unlike previous variants of OT that involve multiple distributions,..." (L19)
3. Although the experimental results demonstrate significant improvements with MLOT over other optimization methods for multiple distributions, a comparison with other optimal transport (OT) methods of similar scope is missing. Specifically, the authors evaluated MLOT-Sinkhorn against non-OT methods, yet did not compare it with other OT-based approaches for finding optimal transport distances between multiple distributions.
4. The overall writing is vague and the flow of the paper is difficult to follow. The related works are non-relevant to the methods proposed. For example, in Section 2, the authors mention Optimal Transport on a Graph and state that "However, the above works rely on the shortest distances on the graph and do not directly compute the transport couplings in the graph. In this paper, we attempt to directly compute the transportation between nodes in a multi-layer structure.". However, the paper lacks further comparisons or references to this aspect in later sections.
5. Typo: In line 259, "As shown in Fig. ,", the figure is not specified.

**Questions:**

1. My biggest question lies in the motivation for finding the OT distance between multiple distributions, which involves uncertain intermediate distributions, and only the source and the target distribution are known. As far as I understand, the motivation of the paper is from real-world problems, where finding the OT distance between two distributions is not enough, the case in which we assume that the source and the target distribution are known.
2. How to compute the intermediate distributions as mentioned in the paper?
3. Can you provide more baselines or related works of optimal transports in literature and explain clearly what is the difference between your methods and these methods?
4. As far as I understand, the authors of (1) propose Sobolev Transport, which aims to find the optimal transport distance between 2 distributions, in which supports of these two distributions are nodes of the graph, which is different from the main purpose of this paper to find optimal transport map between multiple distributions. What does it mean by "we attempt to directly compute the transportation between nodes in a multi-layer structure." (L135-136) since in (1), the nodes of one graph are support of two distributions?

**Remark**: I gave a score of 5, but I am open to raising it once the above concerns are addressed and I have a clearer understanding of the paper. I will also reconsider my score following the rebuttal to ensure my evaluation is fully justified.

(1) Tam Le, Truyen Nguyen, Dinh Phung, and Viet Anh Nguyen. Sobolev transport: A scalable metric for probability measures with graph metrics. In International Conference on Artificial Intelligence and Statistics, pp.9844–9868. PMLR, 2022.

---

> ### Comment · Reviewer_Prsw · 2024-11-26
>
> Dear Authors,
>
> Since the discussion phase has been extended and is now approaching its conclusion in the next eight days, I wanted to kindly follow up as I have not received any responses from you since the rebuttal period began.
>
> Therefore, please submit your rebuttal at the earliest opportunity, allowing me sufficient time to review and provide my responses before the deadline.
>
> Best regards,

---

> ### Author Response · Authors · 2024-11-27
> **Part 1 of Response to Reviewer Prsw**
>
> We greatly appreciate your detailed review and the time you dedicated to evaluating our work. We have carefully considered your feedback and hope the following responses address your concerns.
>
> >***Q1: My biggest question lies in the motivation for finding the OT distance between multiple distributions, which involves uncertain intermediate distributions, and only the source and the target distribution are known. And finding the OT distance between two distributions is not enough***
>
> A1: Thank you for the comments. The motivation behind MLOT comes from real-world problems, specifically cross-border e-commerce operations. In such scenarios, the source distribution (e.g., production capacity) and target distribution (e.g., market demand) are fixed and known, but the intermediate distributions—such as the transportation routes and transit stations—are uncertain and flexible. This uncertainty in the intermediate process inspired us to propose the MLOT problem, which captures the dynamic nature of transport processes that cannot be modeled by traditional optimal transport approaches with fixed marginals.
>
> "Finding the OT distance between two distributions is not enough". You're right. That's why in our approach we not only compute the optimal transport distance but also derive the sequence of optimal transport couplings $(\mathbf{P}_k)_k$ between the source and target distributions. What's more, the intermediate distributions $(\mathbf{a}_k)_k$ at the optimal transportation are computed by the way.
>
> >***Q2: The related works section doesn't mention any previous works on multiple distributions in optimal transport.***
>
> A2: We have discussed the topic of Optimal Transport with multiple marginals in the related work section (Line 113) in our submission. Previous works on multiple distributions, such as Multi-Marginal Optimal Transport(MMOT), was formulated as ***Eq. 3 (Line 116)***: $\min_{P} <\mathbf{C},\mathbf{P}>$, where $\mathbf{C},\mathbf{P}\in U((\mathbf{a}_k)_k)$ and $<\cdot,\cdot>$ represents element-wise product.
>
> Our proposed Multi-Layered Optimal Transport (MLOT) approach introduces two key differences:
> * In MMOT, all marginal distributions are deterministic, whereas in MLOT, **only the source and target distributions are pre-defined. The intermediate distributions are dynamically computed** as part of the optimization process.
> * **MMOT calculates a single coupling** $\mathbf{P}\in {\mathbb{R}^+_{n_1\times n_2\dots n_K}}$ which simultaneously couples all distributions. In contrast, **MLOT computes a sequence of coupling matrices** $\mathbf{P}\in U((\mathbf{a}_k)_k)$
>
> > ***Q3: Comparison with other optimal transport (OT) methods of similar scope is missing. Specifically, the authors evaluated MLOT-Sinkhorn against non-OT methods, yet did not compare it with other OT-based approaches for finding optimal transport distances between multiple distributions.***
>
>
> A3: Thanks for your comment. Please refer to our reply to ***Q6***. In this paper, we compared our method with three OT-based methods in three experiments respectively (Graph OT, CLIP with traditional OT, and Barycenter). We apologize for not expressing this more clearly in the paper, and we will improve the clarity of our expression in the paper.
>
> >***Q4: How to compute the intermediate distributions as mentioned in the paper?***
>
> A4: In our paper, the intermediate distributions are denoted as $(\mathbf{a}_k)_k$, where $k=2,3,\dots,K-1$ (The source and target distributions $\mathbf{a}_1$ and $\mathbf{a}_K$ are given as inputs). These intermediate distributions are iteratively computed using ***Eq. 9 (Line 206)***:
> ```
> $$\mathbf{a}_k=\left\{
> \begin{array}{cc}
>       (\mathbf{u}_k\odot \mathbf{v}_{k-1})^{-\epsilon/\tau} &  \tau>0\\
>      \big((\mathbf{S}_{k-1}^\top\mathbf{u}_{k-1})\odot (\mathbf{S}_{k}\mathbf{v}_{k})\big)^{1/2} & \tau=0
> \end{array}
> \right.$$
> ```
> Upon convergence of the algorithm, the sequence $\mathbf{a_k}$ returned by the algorithm represents the intermediate distributions at the optimal transportation.
> (Or equivalently, $a_k = P_k \mathbf{1} = P_{k-1}^\top \mathbf{1}, ~k=2,\dots,K-1$ after convergence.)

---

> > ### Author Response · Authors · 2024-11-27
> > **Part 2 of Response to Reviewer Prsw**
> >
> > >***Q5: In Section 2, the authors mention "However, the above works rely on the shortest distances on the graph and do not directly compute the transport couplings in the graph. In this paper, we attempt to directly compute the transportation between nodes in a multi-layer structure.". However, The paper lacks further comparisons or references to some related works in later sections.***
> >
> > A5: Thanks you for pointing out. Previous works of OT on graph **relies on computing the shortest path between nodes, and cannot handle constaints on transport flow.** Our MLOT introduces two key differences:
> >
> > * **No need for computing shortest path on graph.** Previous works depend on shortest path computations firstly, while MLOT-Sinkhorn computes the optimal transport flow **directly using the cost metric** in a multi-layered graph. This leads to significant advantages in computational complexity.
> >     * This is illustrated in previous paper, ***Table 1***. The "Short Path+Sinkhorn" method in ***Table 1*** is equivalent to traditonal OT-based approach. Here's the reason: **In a multi-layered graph, once the shortest paths are computed, the problem reduces to standard OT between two distributions,** which can be computed by traditional Ot-Sinkorn. ***Table 1*** **show that MLOT outperforms OT-based methods in terms of speed, maintaining high accuracy as well.**
> > * **Naturally handle constraints on intermediates' distribution.** Previous methods do not account for the computation of intermediate flow, which makes them unsuitable for tasks that involve constraints on intermediate flows (For example, in real-world scenario, each ports may have upper-bound capacity. We may not able to transport much flow through single port). In contrast, our method can naturally handle such constraints. For instance, consider constraints like $\mathbf{a}_k \leqslant \mathbf{s}_k,\forall k=2,3,\dots,K-1$, our algorithm can easily incorporate this by adjusting the flow in each iteration: as long as $\mathbf{a}_k$ is computed during each iteration, we update it by $\mathbf{a}_k = \min(\mathbf{a}_k, \mathbf{s}_k)$.
> >     * To demonstrate our algorithm's effectiveness on problem with constraints, we have conducted additional synthetic experiments, shown in papers' ***Appendix. F Table. 6***. We randomly generate flow constraints for each layer (To increase the difficulty, the sum of the capacity constraints at each layer was set to 1, equaling to total transportation mass). Despite additional constraints, MLOT-Sinkhorn still provides highly accurate results compared to Gurobi, while maintaining efficient computation times.
> >
> > >***Q7: What does it mean by "we attempt to directly compute the transportation between nodes in a multi-layer structure." (L135-136) since in Sobolev Transport, the nodes of one graph are support of two distributions?***
> >
> >
> > A7: We emphasize "directly compute the transportation between nodes in a multi-layer structure." for following three aspects:
> >
> > * Traditional OT operates on a bipartite graph, and Sobolev Transport generalizes this to a graph with arbitrary structure. In contrast, our MLOT lies between these two, as it operates on a multi-layered structured graph.
> > * Sobolev Transport relies on computing the shortest path between nodes. In contrast, our approach does not require shortest path to determine the optimal transport. Instead, we compute the transportation **directly based on the ground metric** $(\mathbf{C}_k)_k$.
> > * Sobolev Transport requires post-processing to determine the flow into each node, which makes handling inequality constraints on flow difficult. In contrast, MLOT **directly computes the flow during the iterative process** and can handle constraints on flow straightforwardly. For instance, if we have a constraint like $\mathbf{a}_k \leqslant \mathbf{s}_k$, this can be easily handled by adding $\mathbf{a}_k = \min(\mathbf{a}_k, \mathbf{s}_k)$ in iteration process.
> >
> > In conclusion, the phrase "directly compute" refers to the fact that, **we can compute the optimal flow as well as intermediate distributions on multi-layered graph directly based on ground metric, no need for shortest path on graph.**
> >
> >
> > >***Q8: About Typo.***
> >
> > A8: Thank you for your feedback. We will make corrections in the final version.

---

> > > ### Author Response · Authors · 2024-11-27
> > > **Part 3 of Response to Reviewer Prsw**
> > >
> > > >***Q6: Can you provide more baselines or related works of optimal transports in literature and explain clearly what is the difference between your methods and these methods?***
> > >
> > > A6: Thanks for your giving us the opportunity for futher explanation. Our experiments involve 5 different baseline (one in supplementary experiment).
> > > * For experiment on synthetic data
> > >
> > >     * **Gurobi**. A commercial solver for general LP problems.
> > >     * **Graph OT**. Graph OT extends the classical OT formulation to problems where $\mathcal{X}$ is a geodesic space$^1$, which means cost metric $c(x,y) = d(x,y)$. When $\mathcal{X}$ is a discrete set, equipped with undirected edges $(i, j) \in \mathcal{E} \subset \mathcal{X}^2$ labeled with a weight $w_{i, j}$, we can recover the geodesic distance (shortest path metric): $D_{i,j} :=\min_{K \geq 0,\{i_m\}: i \rightarrow j} \sum_{k=1}^{K-1} w_{i_k, i_{k+1}}$, where $(i_k, i_{k+1}) \in \mathcal{E}$ and $i \rightarrow j$ means a path from $i$ to $j$. Once $D_{i,j}$ is recovered, cost metric $c(x,y)$ is obtained. Then the coupling $\mathbf{P}$ or the $W_1$ distance between $(\mathbf{a},\mathbf{b})\in(\mathbb{R}^n)^2$ can be sovled by minimizing $\sum_{(i,j)\in\mathcal{E}} c_{i,j}s_{i,j}$, satisfying $\text{div}(s)=\mathbf{a}-\mathbf{b}$
> > >
> > > * For experiment on zero-shot inference for Text-Image retrieval (I would like to emphasize again that **our approach is an training-free improvement**. We reformulate the matching process in the inference stage as a multi-layered graph matching problem.)
> > >     * **Inference by Softmax**. Traditional way to handle similarity matrix. We use $\arg\max \text{softmax}$ for final prediction.
> > >     * **Inference by Traditional OT**. Following OT-CLIP ICML24$^2$, which introduce graph structure and perform graph matching for classification/matching, we conducted new experiment using single OT coupling for prediction, displaying in the darker row in ***Appendix. F Table 3***. This baseline uses single OT coupling for prediction. **The biggest difference with our MLOT lies in the fact that, MLOT leverages all the coupling information (origin data and augmented data), which gains significant performance improvement, while using single OT coupling prediction does not outperform the softmax-based approach.**
> > > * For experiment on computing intermediate images.
> > >     * **Barycenter**$^3$. Given input histogram $\{\mathbf{b_s}\}, s=1,\dots,S$ and weights $\lambda \in \Sigma_S$, a Wasserstein barycenter is computed by minimizing $\sum_{s=1}^S \lambda_s \mathrm{~L}_{\mathbf{C}_s}\left(\mathbf{a}, \mathbf{b}_s\right)$. In the task of computing intermediate images between two given pictures, thus $S=2$ and $\mathbf{C}_s = \mathbf{D}^p\in \mathbb{R}^{n \times n}$ is set to be a distance matrix. Computing the intermediate image is equivalent to find the distribution $\mathbf{a}$ that minimizes: $\lambda_1 W_p^p(\mathbf{a}, \mathbf{b}_1) + \lambda_2 W_p^p(\mathbf{a}, \mathbf{b}_2)$.
> > >
> > >     The comparison between barycenter and MLOT for generating intermediate images is presented in the ***Appendix. F Figure 8***. It's true that both methods perform well in completing the task. However, when $K$ intermediate images are needed to compute, the key advantage of our algorithm lies in the fact that, **MLOT can generate all $K$ intermediate images in a single procedure, while barycenter requires adjusting the weight $K$ times and computing $K$ separate barycenters.** The table below demonstrates the significant time advantage of our algorithm when generating multiple intermediate images.
> > >     |  K  | Barycenter(Second) | MLOT(Second) |
> > >     |:---:|:------------------:|:------------:|
> > >     |  3  |       12.81        |    12.79     |
> > >     |  4  |       25.59        |    19.36     |
> > >     |  5  |       38.67        |    25.26     |
> > >     |  6  |       51.12        |    31.60     |
> > >     |  8  |       77.15        |    43.24     |
> > >     | 10  |       102.21       |    53.83     |
> > > ---
> > > 1. Feldman, Mikhail, and McCann, Robert J. "Uniqueness and Transport Density in Monge's Mass Transportation Problem." Calculus of Variations and Partial Differential Equations, vol. 15, 2002, pp. 81-113.
> > > 2. Shi, Liangliang, Fan, Jack, and Yan, Junchi. "OT-CLIP: Understanding and Generalizing CLIP via Optimal Transport." ICML, 21-27 Jul 2024
> > > 3. J.Rabin,G.Peyr´e, J.Delon, andM.Bernot. Wasserstein barycenter and its application to texture mixing. In A.M. Bruckstein, B.M. ter Haar Romeny, A.M. Bronstein, and M.M. Bronstein(eds.), Scale Space and Variational Methods in Computer Vision, volume 6667 of Lecture Notes in Computer Science.Springer, Berlin, Heidelberg, 2012.

---

### Official Review · Reviewer_XoVS · 2024-11-04

**Soundness:** 2
**Presentation:** 3
**Contribution:** 2
**Rating:** 3
**Confidence:** 4

**Summary:**

The work proposed multi-step optimal transport when finding the OT mapping from one distribution to other distribution, given the supports of "transitional" distributions are known (cost matrices are known). The method is presented in Section 3 with entropic regularization that leads to adapted version of Sinkhorn algorithm to solve the problem. Proposition 3 prove the convergence property of the method. Real applications are text-image retrieval task and generating intermediate images.

**Strengths:**

Empirical results shown in text-image retrieval task, although the setting is quite simple, by creating layers based on flipping the image.
I wonder if there is another work that use augmented data to do this task.

**Weaknesses:**

The theory part is just straight adaptation from known work of Sinkhorn algorithm. In proposition 4, convergence rate depends on $\gamma$, which is not easy to quantify.

For task of generating intermediate images, there is no baselines, or quantity to compare/assess the current method with others. This contribution is limited.

It is not clear (theoretically, intuitively) that in text-image retrieval task, why do we need to use data augmentation to create another layer for OT?

**Questions:**

Cost metric in the task text-image retrieval  could be negative, since it is defined as the cosine similarity, does this affect the definition and algorithm of and using OT?

In both real applications, is there any empirical rule to choose parameter?

---

> ### Author Response · Authors · 2024-11-27
> **Part 1 of Response to Reviewer XoVS**
>
> We appreciate your valuable time and comments and we hope the following answers can address your concerns.
>
> >***Q1: The theory part is just straight adaptation from known work of Sinkhorn algorithm. In proposition 4, convergence rate depends on γ, which is not easy to quantify.***
>
> A1: Thank you for pointing this out. At present, we have proven the global convergence of the algorithm, but a detailed analysis of the overall complexity remains as a future work.
>
> >***Q2: For task of generating intermediate images, there is no baselines, or quantity to compare/assess the current method with others. This contribution is limited.***
>
> A2: Thank you for your comment. We present experimental results comparing the traditional approach of computing intermediate images via **barycenter** with our method using **MLOT**. As shown in the results, MLOT successfully performs this task and achieves slight improvements in smoothness.
>
> However, the main contribution of our work is that, **MLOT allows for the simultaneous computation of all intermediate images in a single step**. In traditional methods via barycenter, it requires repeated computations for each intermediate image with varying $\lambda$ (For each image, we need set cost metric as $[\lambda D,(1-\lambda)D]$). In contrast, MLOT can solve all intermediates by setting cost metric as $[D,D\dots,D]$
>
> |  K  | Barycenter(Second) | MLOT(Second) |
> |:---:|:------------------:|:------------:|
> |  3  |       12.81        |    12.79     |
> |  4  |       25.59        |    19.36     |
> |  5  |       38.67        |    25.26     |
> |  6  |       51.12        |    31.60     |
> |  8  |       77.15        |    43.24     |
> | 10  |       102.21       |    53.83     |
>
> The table below shows the time differences between the two methods for generating varying numbers of intermediate images. When $K=3$ (only 1 intermediate image needs to be generated), both methods demonstrate equivalent time complexity. However, when $K>3$, which requires generating more than one intermediate image, **the time advantage of MLOT becomes significant.**
>
> >***Q3: It is not clear (theoretically, intuitively) that in text-image retrieval task, why do we need to use data augmentation to create another layer for OT?***
>
> A3: We apologize for not providing a clearer explanation. In previous works on CLIP, inference typically involves computing the predicted probabilities using softmax, then selects the most likely class with $\arg\max$.
>
> Following the REOT NIPS23$^1$, classification (or matching) tasks can be viewed as a special case of OT. **By introducing augmented data, we extend this view into the framework of Multi-layered OT**.
>
> What's more important is that, our method **introduces data augmentation during inference**. In contrasts, other augmentation-based methods, such as MixGen$^2$, CLIP-MedFake$^3$ etc., **requires retraining to improve CLIP's performance.** Our approach, by leveraging additional information, treats retrieval as a MLOT problem, and it's a **training-free improvement.**
>
> As shown in figure below, we record the respective prediction results from 2 MLOT coulings (Since 3 layers in total) separately. As can be seen, **using individual OT predictions alone does not significantly improve accuracy; only when both are used in MLOT do we observe a noticeable improvement.**
>
> ---
> 1. Shi, Liangliang, Zhen, Haoyu, Zhang, Gu, and Yan, Junchi. "Relative Entropic Optimal Transport: A (Prior-aware) Matching Perspective to (Unbalanced) Classification." Advances in Neural Information Processing Systems (NeurIPS), 2023.
> 2. https://openaccess.thecvf.com/content/WACV2023W/Pretrain/papers/Hao_MixGen_A_New_Multi-Modal_Data_Augmentation_WACVW_2023_paper.pdf
> 3. H. Chen et al., "Clip-Medfake: Synthetic Data Augmentation With AI-Generated Content for Improved Medical Image Classification," 2024 IEEE International Conference on Image Processing

---

> > ### Author Response · Authors · 2024-11-27
> > **Part 2 of Response to Reviewer XoVS**
> >
> > >***Q4: Cost metric in the task text-image retrieval could be negative, since it is defined as the cosine similarity, does this affect the definition and algorithm of and using OT?***
> >
> > A4: Thanks for pointing this out. In experiment, we modify the cost metric in our MLOT-Sinkhorn algorithm by adding a constant to ensure that the cost matrix is non-negative (e.g., if cosine similarity is $\mathbf{S}_k$, then we use $(\text{Const}-\mathbf{S}_k)$ as cost metric $\mathbf{C}_k$. This approach is similar to the one used in InverseOT ICML24$^4$.
> >
> > >***Q5: In both real applications, is there any empirical rule to choose parameter?***
> >
> > A5: In practical applications, when runtime is not a primary concern, setting $\tau=0$ is often the best choice.
> >
> > As work InverseOT ICML24$^4$ shows, the parameter $\varepsilon$ is effectively equivalent to the temperature in the softmax function, so tuning $\varepsilon$ in our algorithm is analogous to adjusting the softmax temperature.
> >
> > For CLIP task, we set $\varepsilon=5\times 10^{-2},\tau=0$. For intermediate image task, we set $\varepsilon=5\times 10^{-5},\tau=5\times 10^{-4}$
> >
> > ---
> > 4. Shi, Liangliang, Fan, Jack, and Yan, Junchi. "OT-CLIP: Understanding and Generalizing CLIP via Optimal Transport." Proceedings of the 41st International Conference on Machine Learning, 21-27 Jul 2024

---

> > > ### Comment · Reviewer_XoVS · 2024-11-28
> > > **Reply to rebuttal**
> > >
> > > I would like to thank authors for their reply. After reading the answers for Q3 and Q4, I still think they are like   techniques rather than intuitive explanation for physical discovery to find out steps in the discrete process minimizing OT distance. I do not think I could raise the score above the threshold, because the theory part is too weak, the application part has only one real application and it does not seem to be the exemplary example of MLOT.

---

### Official Review · Reviewer_MXuv · 2024-11-11

**Soundness:** 3
**Presentation:** 2
**Contribution:** 2
**Rating:** 3
**Confidence:** 4

**Summary:**

The authors propose to discretize the path of intermediate distributions between source and target distributions in the OT problem. They added up the total cost between each two adjacent transportation problems on the path and adopted the Sinkhorm iterations to alternatively solve for the intermediates and the couplings which ensemble the transportation plan. Convergence was provided and experiments showed performance gains in solving the text-image retrieval problem.

**Strengths:**

The authors wrapped their MLOT formation with the Sinkhorn algorithm and thus inherited all the computational benefits of it. Theoretical results seem solid to me. Convergence analysis was provided and verified with synthetic data.

**Weaknesses:**

The main problem I find in the paper is that the authors didn't answer the very question that they raised. In the second paragraph (line 039), they referenced transportation as the "real-world" example of the benefit of MLOT, by optimizing the freight transportation to reduce the cost and environmental impact, as suggested by their reference Bektas et al. 2019. However, throughout the paper, they didn't answer the question. Can MLOT help find the optimal routes and minimize the total cost?

In real-world problems, the intermediate distributions have certain constraints like the routes (cost matrix) and max capacity of the stops (intermediate distributions) in freight transportation example. The paper doesn't mention such constraints either in methodology or experiments.

In 4.2, MLOT was compared against softmax so it's not clear whether the gains were from OT or from the multi-layer formulation or both.

Figure 7 in 4.3 also demonstrated that intermediate distributions may not be physically meaningful and smooth and thus not applicable to solving "real-world problems".

**Questions:**

Line 323: Therefore, out MLOT can offer new perspectives and approximate computations for Dynamic OT and the Schrodinger bridge. What are the new perspectives? That MLOT can "approximate computations for Dynamic OT and the Schrodinger bridge" is a serious claim. I'd like the authors to provide evidence for that or tune down the sentence to something like whether we can approximate dynamic OT via MLOT is a future direction. And it's not clear in what aspect do we approximate. Is it the total cost or the transportation map or something else.

Comparing MLOT with barycenters should be expanded. In line 182, "when $S=2$ in Eq.6 and $K=3$ in Eq.5, the optimization of our MLOT is equivalent to solving the Wasserstein barycenter." It's not, because for that to be true you have to assume $\lambda = \frac{1}{S} = \frac{1}{2}$, don't you? As we change $\lambda$ in barycenters, we could also form a path of intermediate distributions between the target and the source. How is that process compared to MLOT?

---

> ### Author Response · Authors · 2024-11-27
> **Part 1 of Response to Reviewer MXuv**
>
> We appreciate your valuable time and comments and we hope the following answers can address your concerns.
>
> >***Q1: The main problem I find in the paper is that the authors didn't answer the very question that they raised. Can MLOT help find the optimal routes and minimize the total cost?***
>
> A1: Thank you for your comment. We apologise for the potential misunderstanding. The real-world scenario (cross-border e-commerce) is the primary *motivation* for proposing our algorithm.
>
> Similar to how traditional OT theory originated from the simple transportation problem (moving goods from sources directly to destinations), our algorithm originates from the more realistic and generalized scenario, since real-world transportation problems often involve intermediate stops, making them inherently multi-layered. Inspired by these complexities, we extend the classical OT framework to multi-layered framework.
>
> As noted by works like Marco Cuturi's introduction of the Sinkhorn algorithm$^1$, which provides an efficient approximation for traditional OT problems, our algorithm similarly offers a computationally efficient approach to solving optimal transport in multi-layered scenarios.
>
>
> >***Q2: In real-world problems, the intermediate distributions have certain constraints like the routes (cost matrix) and max capacity of the stops (intermediate distributions) in freight transportation example. The paper doesn't mention such constraints either in methodology or experiments.***
>
> A2: Thank you for your insightful suggestion. Our algorithm is designed to be flexible and can naturally handle such constraints.
>
> For instance, when dealing with capacity constraints on the intermediate nodes, i.e. given constraints $\mathbf{s}_k$ on intermediates' distribution, $\mathbf{a}_k\leqslant\mathbf{s}_k,~k=2,3\dots K-1$. Our algorithm can easily incorporate this by adjusting the flow in each iteration: as long as $\mathbf{a}_k$ is computed during each iteration, we update it by $\mathbf{a}_k = \min(\mathbf{a}_k, \mathbf{s}_k)$.
>
> To demonstrate our algorithm's effectiveness, we have conducted additional synthetic experiments, shown in papers' ***Appendix. F Table. 6***. We randomly generate flow constraints for each layer (To increase the difficulty, the sum of the capacity constraints at each layer was set to 1, equaling to total transportation mass). Despite additional constraints, MLOT-Sinkhorn still provides highly accurate results compared to Gurobi, while maintaining efficient computation times.
>
> >***Q3: In 4.2, MLOT was compared against softmax so it's not clear whether the gains were from OT or from the multi-layer formulation or both.***
>
>
> A3: Thanks for your giving us the opportunity for futher explanation. Our experiments are **purely inference-based, with no retraining involved**. The model used is the standard CLIP model$^2$.
>
> Following the REOT NIPS23$^3$, classification (or matching) tasks can be viewed as a special case of OT. **By introducing augmented data, we extend this view into the framework of Multi-layered OT**.
>
> What's more important is that, our method introduces data augmentation **during inference**. In contrast, other augmentation-based methods, such as MixGen$^4$, CLIP-MedFake$^5$ etc., **requires retraining to improve CLIP's performance.** Our approach, by leveraging additional information, treats retrieval as a MLOT problem, and it's a **training-free improvement.**
>
>
> We conduct a new experiment, recording the respective prediction results from 2 MLOT coulings (Since 3 layers in total) separately, shown in ***Appendix. F Table. 3***. As can be seen, **using individual OT predictions alone does not significantly improve accuracy; only when both are used in MLOT do we observe a noticeable improvement.**
>
> ---
> 1. Cuturi, Marco. "Sinkhorn Distances: Lightspeed Computation of Optimal Transport." Advances in Neural Information Processing Systems, vol. 26, 2013.
> 2. https://github.com/mlfoundations/open_clip
> 3. Shi, Liangliang, Zhen, Haoyu, Zhang, Gu, and Yan, Junchi. "Relative Entropic Optimal Transport: A (Prior-aware) Matching Perspective to (Unbalanced) Classification." Advances in Neural Information Processing Systems (NeurIPS), 2023.
> 4. https://openaccess.thecvf.com/content/WACV2023W/Pretrain/papers/Hao_MixGen_A_New_Multi-Modal_Data_Augmentation_WACVW_2023_paper.pdf
> 5. H. Chen et al., "Clip-Medfake: Synthetic Data Augmentation With AI-Generated Content for Improved Medical Image Classification," 2024 IEEE International Conference on Image Processing

---

> > ### Author Response · Authors · 2024-11-27
> > **Part 2 of Response to Reviewer MXuv**
> >
> > > ***Q4: Figure 7 in 4.3 also demonstrated that intermediate distributions may not be physically meaningful and smooth and thus not applicable to solving "real-world problems"***
> >
> > A4: Thank you for your comment. The main purpose of this paper is not to directly and completely solve real-world problems, but rather to propose a generalization of traditional OT, inspired by real-world settings that canot be addressed by vanilla OT.
> >
> > The experiment in Section 4.3 was inspired by barycenter computation. In traditional methods, if we want to generate $N$ intermediate images, we must set different $\lambda$ values and repeat computing the barycenter $N$ times.
> >
> > In contrast, the key contribution of our work is that **MLOT allows for the simultaneous computation of all intermediate images in a single step.**
> >
> > The following table shows wall-clock time cost by different methods computing intermediate images. The results show significant speed improvement, especially in larger $K$ (more intermediate images needs to be computed).
> >
> > |  K  | Barycenter(Second) | MLOT(Second) |
> > |:---:|:------------------:|:------------:|
> > |  3  |       12.81        |    12.79     |
> > |  4  |       25.59        |    19.36     |
> > |  5  |       38.67        |    25.26     |
> > |  6  |       51.12        |    31.60     |
> > |  8  |       77.15        |    43.24     |
> > | 10  |       102.21       |    53.83     |
> >
> >
> > >***Q5: That MLOT can "approximate computations for Dynamic OT and the Schrodinger bridge" is a serious claim. I'd like the authors to provide evidence for that. And it's not clear in what aspect do we approximate. Is it the total cost or the transportation map or something else.***
> >
> > A5: Thanks for your comments. We have updated detailed discussion in ***Appendix. G.***
> >  Solving the full Schrodinger Bridge problem, especially in its continuous form, can be computationally difficult. Several numerical methods were proposed, such as IPF$^6$, DifussionSB$^7$.
> >
> > Under mild assumptions, the static SB problem can be seen as an entropy-regularized optimal transport problem: $$\min -\mathbb{E}\left[\log p_{N \mid 0}\left(X_N \mid X_0\right)\right]-\mathrm{H}\left(\pi^{\mathrm{s}}\right)$$
> >
> > As a comparison, the KL form of MLOT problem (presented in ***Eq.11***): $$\min \varepsilon \sum \widetilde{KL}(\mathbf{P}_k|\mathbf{S}_k) + \tau \sum  \widetilde{KL}  (\mathbf{a}_k |\Delta_k)$$
> >
> > As proved in ***Appendix.C***, minimizing the part $\tau{KL}(\mathbf{a}_k | \Delta_k)$ is equivalent to minimizing $H(\mathbf{a}_k)$. Therefore, our MLOT problem shares a **similar KL divergence structure to the discrete form of the Schrödinger bridge.**
> >
> >
> > By drawing this parallel, we suggest that in the special case where SB problem is discrete, MLOT-Sinkhorn provides a potential approach to solving the SB problem.
> >
> > >***Q6: As we change $\lambda$ in barycenters, we could also form a path of intermediate distributions between the target and the source. How is that process compared to MLOT?***
> >
> > A6: our method is equivalent to barycenter when $K=3$, as long as we set the cost metric in MLOT to $\mathbf{C}_1=\lambda D,~\mathbf{C}_2=(1-\lambda) D$, which corresponds to the different $\lambda$ used in the barycenter.
> >
> > However, the key difference lies in the efficiency of the two approaches. **In scenarios where multiple intermediate images are needed, barycenter requires re-calculating for each $\mathbf{\lambda}$ value, In contrast, our MLOT algorithm computes all intermediate images in one procedure.**
> >
> > ---
> > 6. Chen, Y., Georgiou, T. T., and Pavon, M. (2021b). Optimal transport in systems and control. Annual Review of Control, Robotics, and Autonomous Systems, 4.
> > 7. De Bortoli, Valentin, Thornton, James, Heng, Jeremy, and Doucet, Arnaud. "Diffusion Schrödinger Bridge with Applications to Score-Based Generative Modeling." 2023. arXiv:2106.01357 [stat.ML]

---

### Author Response · Authors · 2024-12-01
**General Response and Summary of Updates to Manuscript**

We sincerely thank the reviewers and AC for investing their valuable time and efforts in this paper.

We thank the reviewers for noting that we introduce a variant of multiple distributions optimal transport and a corresponding variant of the Sinkhorn algorithm to solve this problem (MXuv, XoVS, Prsw, b2Pr), with solid (MXuv) theoretical analysis on the convergence property (XoVS, b2Pr), as well as experiments that demonstrates superior efficiency on complex problems while maintaining accuracy (Prsw, b2Pr). The experimental section is thorough and clearly written (Prsw), and the paper is easy to follow (b2Pr).

First, we provide a high-level summary of the changes that we've made to the draft to address your feedback, and conclude with an overview of our key contributions, and how they differ from previous work.

**Summary of changes made to the draft**
* Revamped **introduction and related work section** to clarify how our MLOT formulation was motivated from real-world scenario, and how MLOT differs from other multi-distribution OT. (**MXuv, Prsw**)
* Added detailed discussion in **Appendix. G** about why MLOT can offer new perspectives and approximate computations for Dynamic OT and Schrödinger Bridge (**MXuv**).
* Added detailed experiment on Retrieval task in **Appendix. F Table 3**, recording the performance using traditional OT. Without formulating the problem as MLOT, it cannot outperform using softmax. (**MXuv, b2Pr**)
* Added additional experiment on Constraint-MLOT problem in **Appendix. F Table 6**, i.e. exists flow constraint on each intermediate distributions. The results demonstrate our MLOT-Sinkhorn algorithm's strong adaptability to such constrainted problem. (**MXuv, Prsw**)
* Added additional experiment on sythetic data with different shape of $(\mathbf{n}_k)_k$ in **Appendix. F Table 5**, to show that our MLOT-Sinkhorn can work on arbitary intermediate shape. (**b2Pr**)
* Added additional experiment result of comparison of two methods computing intermediate images: via Barycenter and via MLOT, in **Appendix. F Figure 8**. And further record their computing time, which shows our simplify in computing procedure and improvement in efficiency. (**Prsw, b2Pr**)

**Novelty and technical insight**

We would like to clarify that our proposed MLOT has different problem setting with previous multiple distribution OT, and ours were motivated by a more complicated real-world scenario compared to traditional OT. Details are summarized in the following table.
||Structure|Known Information|Transport Result|Handle Constraints while Maintaining Original Speed|Requirement for Shortest Path|
|-|-|-|-|-|-|
|**Traditional OT**|Bipartite distributions|Ground metric, source & target distributions|One $\mathbf{P}$, couples source and target|Yes|-|
|**Multi-Marginal OT**|Multiple layered graph, with multiple marginal distributions|Ground metric,all marginals' distributions|One coupling, simultaneously couples all distributions|Yes|No need|
|**Graph OT**|Arbitrary graph|Ground metric, original & final distributions|Flow $s$, transfers original distribution to final|No|Need|
|**Multi-Layered OT**|Multiple layered graph, with multiple marginal distributions|Ground metric, source & target distributions, **not include intermediates'**|A sequnce of couplings, couples each pair of adjacent distributions|Yes|No need|

We now summarize our key contributions:

1. **Algorithm solving MLOT with high efficiency and accuracy.** We propose MLOT-Sinkhorn, **prove its global convergence, verify its convergence, accuracy and high efficiency by a wide range of sythetic experiment**. Compared to previous OT-based method for solving MLOT, which requires firstly computing shortest path on graph based on ground metric. In contrast, our algorithm **compute optimal flow directly by ground metric**. What's more, our algorithm can **naturally adapt the constrainted-MLOT**, which Graph OT cannot handle.
2. **A Training-Free improvment for Zero-shot Image-Text Retrieval task.** Following OT-CLIP^[ICML24], which introduce graph structure and perform graph matching for classification/matching, we **reformulate the image-text retrieval task as MLOT problem**, by introducing data augmentation. MLOT can leverage two coupling's information, outperform methods using softmax or traditional OT. What's more important is that, differing from other works using data augmentation technique, our method is a **training-free improvement.**
3. **A new perspective to computing intermediate images.** When source and target in MLOT are each an image, the intermediate distributions will exactly be intermediate images. Traditionally, we compute barycenter for this task. The drawback of barycenter is that when multiple intermediate images are need, we have to compute several barycenter respectively. In contrast, **our method has significant time efficiency, which only requires a single computation process and returns all intermediate images.**

---

### Meta-Review · Area_Chair_ufge · 2024-12-21

**Metareview:**

In the paper, the authors proposed multi-layered optimal transport (OT) framework that generalizes the traditional two-layer OT structure to handle transportation challenges across multiple hierarchical levels. For the practical computation, the authors also proposed entropic regularization of the multi-layered OT in which the Sinkhorn can be used and accelerated using GPUs. Finally, convergence results are also provided and are verified under the synthetic settings.

After the rebuttal, there are still several weaknesses of the current paper: (1) The theoretical results are not novel, especially those with the Sinkhorn algorithm, as they are adapted directly from the known results. (2) The experiments lack baselines and are not extensive. It is not clear if the proposed method can outperform methods with similar scope. (3) The current writing of the paper is generally hard to follow. Several parts are not closely connected. (4) Finally, the motivation of the current paper is not convincing.

Given the above weaknesses of the paper, I recommend rejecting the paper at the current stage. The authors are encouraged to incorporate the suggestions and feedback of the reviewers into the revision of their manuscript.

**Additional Comments On Reviewer Discussion:**

Please refer to the meta-review.

---

### Decision · Program_Chairs · 2025-01-22

Reject